# TEXT-TO-LORA: INSTANT TRANSFORMER ADAPTION

## ABSTRACT

While Foundation Models provide a general tool for rapid content creation, they regularly require task-specific adaptation. Traditionally, this exercise involves careful curation of datasets and repeated fine-tuning of the underlying model. Fine-tuning techniques enable practitioners to adapt foundation models for many new applications but require expensive and lengthy training while being notably sensitive to hyper-parameter choices. To overcome these limitations, we introduce Text-to-LoRA (**T2L**), a model capable of adapting Large Language Models *on the fly* solely based on a natural language description of the target task. **T2L** is a hypernetwork trained to construct LoRAs in a single inexpensive forward pass. After training **T2L** on a suite of 9 pre-trained LoRA adapters (GSM8K, Arc, etc.), we show that the ad-hoc reconstructed LoRA instances match the performance of task-specific adapters across the corresponding test sets. Furthermore, **T2L** can compress hundreds of LoRA instances and zero-shot generalize to entirely unseen tasks. This approach provides a significant step towards democratizing the specialization of foundation models and enables language-based adaptation with minimal compute requirements. Our code and pre-trained checkpoints will be available through GitHub and HuggingFace upon publication.

## 1 INTRODUCTION

Biological systems are capable of rapid adaptation, given limited sensory cues. For example, the human visual system can tune its light sensitivity and focus through neuromodulation of the fovea and rod cells (Wurtz et al., 2011; Digre & Brennan, 2012). While recent Large language models (LLMs) exhibit a wide variety of capabilities and knowledge, they remain rigid when adding task-specific capabilities. In such cases, practitioners often resort to re-training parts of the model (Gururangan et al., 2020; Wei et al., 2021; Dettmers et al., 2022; Tay et al., 2021) using parameter-efficient fine-tuning techniques, e.g., Low-Rank Adaptation (LoRA, Hu et al., 2022). Typically, a LoRA adapter has to be optimized for each downstream task and requires task-specific dataset and hyperparameter setting. This fine-tuning scheme for adaptation significantly limits the possibility of transferring knowledge between tasks and induces engineering overhead.

Recently, it has been observed that by inducing structural constraints, the low-rank matrices learned by LoRA adapters can be further compressed. For example, one can train *lossy* versions of the original adapter while maintaining downstream performance (Brüel-Gabrielsson et al., 2024; Kim et al., 2024; Kopiczko et al., 2024). Furthermore, multiple LoRAs can be combined for new tasks at inference time (Ostapenko et al., 2024). At the core of these approaches lies the explicit use of decomposition or dimensionality reduction techniques (e.g., SVD or routing) for better compression and online composition of existing LoRAs. This raises the following questions:

> 1. Can we end-to-end train a neural network to compress many pre-trained LoRAs?
>
> 2. Can we decode new task-specific LoRA adapters from this network solely based on natural-language instructions for an unseen task at test time?

We hypothesize that different LoRA adapters share the same underlying adaptation mechanism and can be optimized simultaneously without any explicit structure or recipe for combining them. To explicitly test this hypothesis, we propose **T2L** (see Section 1), a hypernetwork (Ha et al., 2016) that compresses task-specific LoRAs and generates new LoRA adapters *zero-shot* at inference time. **T2L**

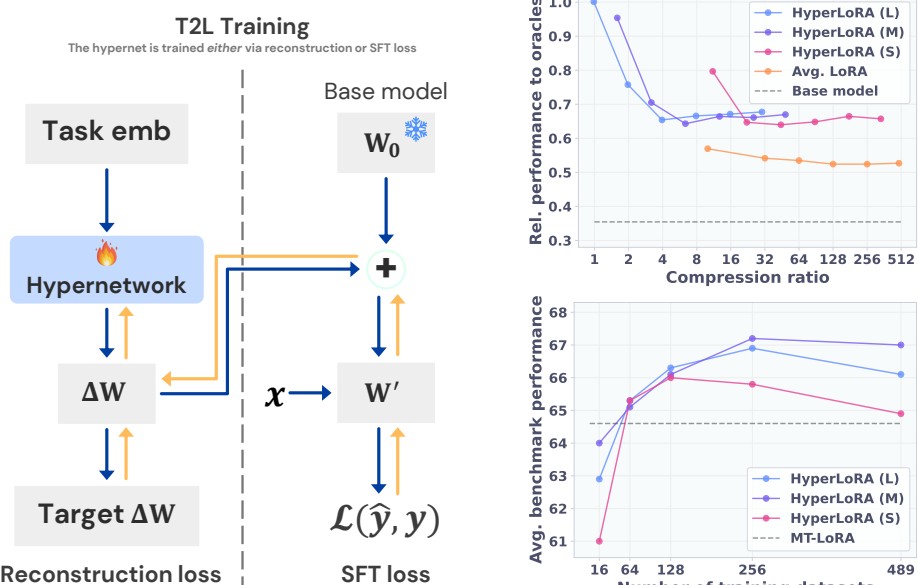

Figure 1: **Left:** Conceptual overview of T2L training routine. Given a set of task description embeddings, we train a hypernetwork to generate LoRA adaptation matrices ($\Delta W$) for various tasks. The weights of T2L are *either* optimized to distill pre-trained LoRA weights or via multi-task supervised fine-tuning on downstream tasks. Blue and Orange arrows refer to forward and backward propagation. **Right, Top:** Relative performance to the oracles on training SNI tasks with varying compression ratios. **Right, Bottom:** Zero-shot LoRA generation performance on 10 benchmark tasks. As we increase the number of pre-training datasets, the performance of T2L increases for three different T2L architectures.

is trained to compress LoRAs on a diverse task distribution from the Super Natural Instruction (SNI) dataset (Wang et al., 2022). Importantly, T2L takes a natural language description of the target task as an input, allowing zero-shot LoRA generation to unseen tasks. Empirically, we show that T2L can effectively be trained either to reconstruct pre-trained adapters or via supervised fine-tuning on distribution of downstream tasks (see Fig. 1, top left). After training, T2L outperforms a multi-task LoRA baseline and Arrow Routing (Ostapenko et al., 2024), a state-of-the-art zero-shot LoRA routing method, on various benchmark tasks. Furthermore, we show that T2L can generate LoRA adapters for previously unseen tasks solely using the language-based task description. This result highlights the generalization capabilities and applicability of our proposed indirect adaptation encoding. Our contributions are summarized as follows:

1. We introduce hypernetwork-based architectures for producing LoRA adapters with a single forward pass (Section 3) based on text descriptions. T2L architectures can be trained using both distillation of pre-trained adapters and supervised multi-task fine-tuning.

2. We show that T2L can efficiently encode hundreds of LoRA adapters (Section 4). While the compression is lossy, T2L maintains the performance of task-specifically tuned LoRA adapters. Furthermore, T2L can generalize to unseen tasks given suitable natural language descriptions of the tasks.

3. We provide rigorous ablations (Section 5) including T2L scaling with datasets (see Fig. 1, bottom right), the impact of different task description embeddings, the training routines, and text-based task descriptions.

4. Finally, we provide various efforts to analyze the nature of T2L generations (Section 6). We study the relationship between LoRA adapters and find compelling evidence why reconstruction-trained T2L cannot generalize. Furthermore, we find semantically meaningful LoRA clusters when visualizing the generated LoRAs in a dimensionality-reduced space.

## 2 PRELIMINARIES

We utilize multiple fine-tuning datasets $\mathcal{D} = \{\mathcal{D}^1, \ldots, \mathcal{D}^T\}$, which correspond to different tasks $\mathcal{T} = \{t^1, \ldots, t^T\}$. For the purpose of training T2L, we assume that each fine-tuning dataset has a set of natural language *task descriptions* ($Z^i = \{z_1^i, \ldots, z_m^i\}$): $\mathcal{D}^i = \{X^i, Y^i, Z^i\}$. The task descriptions do not need to be specific to each sample but rather a general description of the dataset. For a single task $t^i$, the fine-tuning objective of an LLM with pre-trained weights ($\Psi$) is given by

$$\Delta \boldsymbol{W}^i = \underset{\Delta \boldsymbol{W}^i}{\arg\min} \, \mathcal{L}_{\text{SFT}}(\mathcal{D}^i, \Psi, \Delta \boldsymbol{W}^i), \tag{1}$$

where $\mathcal{L}_{\text{SFT}}$ gives the supervised fine-tuning loss and $\Delta \boldsymbol{W}^i$ is the fine-tuning adaption for task $t^i$ to the base weights. For the *multi-task* setting, we train a single adapter $\Delta \boldsymbol{W}$ to minimize the expected loss over the union of all datasets $\mathcal{D}$:

$$\Delta \boldsymbol{W} = \underset{\Delta \boldsymbol{W}}{\arg\min} \, \mathbb{E}_{\mathcal{D}^i \sim \mathcal{D}} \, \mathcal{L}_{\text{SFT}}(\mathcal{D}^i, \Psi, \Delta \boldsymbol{W}). \tag{2}$$

**Low-Rank Adaptation (LoRA, Hu et al., 2022):** LoRA is a parameter-efficient fine-tuning method that freezes the pre-trained weights of a base model and only learns low-rank weight matrices, which serve as an adapter to the base model. For each selected linear transformation $h = \boldsymbol{W}_0 \boldsymbol{x}$, the fine-tuned transformation is given by $h = \boldsymbol{W}_0 \boldsymbol{x} + \Delta \boldsymbol{W} \boldsymbol{x} = \boldsymbol{W}_0 \boldsymbol{x} + \boldsymbol{B}^T \boldsymbol{A} \boldsymbol{x}$, where $\boldsymbol{A}, \boldsymbol{B} \in \mathbb{R}^{r \times d}$ are weight matrices of rank $r < d$. We drop the layer index and module type of the LoRA weights when referring to all LoRA weights simultaneously. Otherwise, we use subscripts to represent the layer index and module type, e.g., $\Delta \boldsymbol{W}_{m,l}$, where $m$ is the module type (e.g., query projection) and $l$ is the layer index.

**Hypernetworks:** A hypernetwork is a neural network that generates parameters for another 'base' network (Ha et al., 2016). It serves as an indirect encoding (Schmidhuber, 1997; Stanley & Miikkulainen, 2003; Zhang et al., 2018; Schug et al., 2024) of the base network, given that the parameter count of the hypernetwork is much smaller. This compression is achieved by learning to share parameters indirectly. More specifically, given a layer-specific descriptor vector $\phi_l$, a hypernetwork with parameters $\theta$ generates the parameters of the base model at layer $l \in \{1, \ldots L\}$ as follows: $\boldsymbol{W}_l = h_\theta(\phi_l)$. Traditionally, the layer descriptors are either one-hot or learned vectors. The weights $\theta$ are then trained via end-to-end optimization on a downstream task.

## 3 TEXT-TO-LORA: LEARNING TO COMPRESS AND GENERATE LORAS

In this work, we utilize a hypernetwork to generate LoRA adapters for task-specific adaptation. For each target module ($m$) and layer index ($l$), a hypernetwork generates the two low-rank matrices $\boldsymbol{A}, \boldsymbol{B}$ based on a task description $z^i \in Z^i$ of a task $t^i$ as follows:

$$\Delta \boldsymbol{W}_{m,l}^i = h_\theta(\phi_{m,l}^i), \text{ with } \phi_{m,l}^i = \mathtt{concat}\left[f(z^i), E[m], E[l]]\right], \tag{3}$$

where $f$ gives a vector representation of a text description, typically represented by a CLS token of a bidirectional transformer model or last token activation of an LLM. $E$ is a learnable embedding dictionary indexed by either a module type $m$ or a layer index $l$. For legibility, we introduce a shorthand notation for T2L's output $\Delta \boldsymbol{W}^i := h_\theta(\phi^i) := h_\theta(\{\phi_{m,l}^i\})$. Then, a supervised fine-tuning training objective for T2L is

$$\theta = \underset{\theta}{\arg\min} \, \mathbb{E}_{\mathcal{D}^i \sim \mathcal{D}, z^i \sim Z^i} \, \mathcal{L}_{\text{SFT}}(\mathcal{D}^i, \Psi, h_\theta(\phi^i)), \tag{4}$$

Note that values of $m$ and $l$ can be batched, which allows T2L to generate $\Delta W$ for all the modules and layer indices efficiently within a single forward pass.

### 3.1 HYPERLORA ARCHITECTURES

Most of a hypernetwork's parameters come from the output layer, which scales linearly with the size of the target weights (Von Oswald et al., 2019). To explore the complexity-performance trade-off, we propose three variants of T2L: L , M , and S . We impose different output spaces on the hypernetwork that represent different inductive biases and parameter counts (see Fig. 2). We note that all variants use the same backbone architecture and only differ in their output heads and

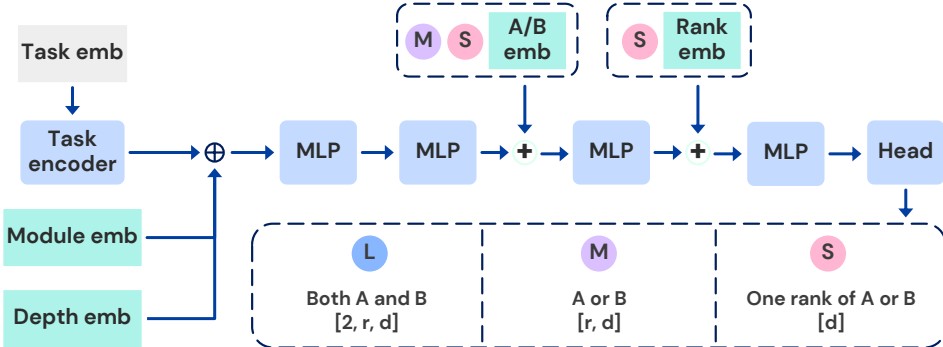

Figure 2: Overview of **T2L** architectural variations. The dashed box at the bottom shows the output size of a single forward pass of **T2L**. Blue boxes are trainable modules. Cyan Boxes are trainable embedding layers. Components in dashed boxes are only used with their corresponding architectures. $r$ is the rank of a LoRA adapter and $d$ is the size of the input and the output dimension.

learnable embeddings. The **L** **architecture** is the largest variant. Its final linear layer outputs low-rank $\boldsymbol{A}$ and $\boldsymbol{B}$ matrices simultaneously with the number of weight connections to the output head $|\theta_{\text{head}}| = d_{\text{out}} \times 2 \times r \times d$, where $d_{\text{out}}$ is the output size of the last MLP block. **M** **architecture** is the medium-size model with a shared output layer between the low-rank $A$ and $B$ matrices. That is, the head outputs a low-rank matrix, either $A$ or $B$, depending on the learnable embedding. The size of the output head is $|\theta_{\text{head}}| = d_{\text{out}} \times r \times d$. Finally, **S** **architecture** is the most parameter-efficient model with the strongest inductive biases, where the hypernetwork outputs only one rank of a low-rank matrix at a time. This output space makes the size of the head much smaller: $|\theta_{\text{head}}| = d_{emb} \times d$. For reference, a LoRA adapter has $r \times d \times 2 \times L \times |M|$ trainable parameters, where $L$ is the number of layers and $|M|$ is the number of target modules. The default value of $d_{\text{out}}$ is 512. We note that every architecture can generate the entirety of low-rank matrices $\boldsymbol{A}$ and $\boldsymbol{B}$ in a single forward pass by batching all the input embeddings. We provide more details of the architectures in Appendix B and the weight initialization method that leads to stable training in Appendix C.

### 3.2 TRAINING HYPERLORA VIA LORA RECONSTRUCTION (DISTILLATION)

The most straightforward way to train **T2L** is to reconstruct pre-trained task-specific LoRAs. This setup allows us to utilize publicly available libraries of LoRAs (Brüel-Gabrielsson et al., 2024; Zhao et al., 2024). Alternatively, one can also use a two-stage procedure, in which a library of LoRAs is pre-trained in the first stage and then train **T2L** to reconstruct them. For the sole purpose of compressing LoRAs, we can train **T2L** using one-hot or learnable vectors as task embeddings. However, these embeddings do not allow zero-shot LoRA generation for unseen tasks. To enable zero-shot LoRA generation, we additionally condition **T2L** with embeddings of natural language task descriptions, which allows **T2L** to generate LoRA adapters for various tasks—including unseen ones—given corresponding task descriptions. Given a suitable library of LoRA adapters $\Omega$, the reconstruction loss for **T2L** can be written as

$$\mathcal{L}(\Omega, \theta) = \mathbb{E}_{\Delta \boldsymbol{W}^i \sim \Omega} \left| \Delta \boldsymbol{W}^i - h_\theta(\phi^i) \right|. \tag{5}$$

### 3.3 TRAINING HYPERLORA VIA SUPERVISED FINE-TUNING

Alternatively, **T2L** can be directly optimized on fine-tuning datasets. Training **T2L** with SFT sidesteps the need for intermediate target LoRA adapters and allows for end-to-end training of the hypernetwork. This training scheme is preferred if existing trained LoRAs are not naturally clustered by their functionalities or downstream tasks. For instance, $t^1$ and $t^2$ could be two related tasks requiring a similar LLM capability, but $\Delta \boldsymbol{W}^1$ and $\Delta \boldsymbol{W}^2$ could be in different minima. Thus, **T2L** trained via reconstruction training would have to compress numerically different $\Delta \boldsymbol{W}^1$ and $\Delta \boldsymbol{W}^2$, making it less likely to generalize. In fact, we empirically find that a **T2L** trained via reconstruction fails to generalize to unseen tasks (Section 5.6). In contrast, an SFT-trained **T2L** can implicitly learn to cluster tasks, which has been shown to improve zero-shot LoRA routing performance (Ostapenko et al., 2024). The SFT loss for **T2L** is given by Eq. (4).

# 4 EXPERIMENTS

We investigate the effectiveness of the different `T2L` architectures and training schemes in terms of the compression of adapters (Section 4.1) and zero-shot LoRA generation for unseen tasks (Section 4.2). As baselines, we consider task-specific LoRAs, element-wise averaged LoRA, and multi-task LoRA— a LoRA adapter trained on all training tasks. We also implement Hyperdecoders (Ivison & Peters, 2022)—a hypernetwork that generates LoRAs on a per-sequence basis—based on our proposed architectures. To boost the performance of the base models without fine-tuning, we utilize few-shot in-context learning (ICL, Brown et al., 2020; Dong et al., 2024) and task description prepending, i.e., providing task description at the beginning of each query. Additionally, we include results of Arrow Routing zero-shot performance from Ostapenko et al. (2024). Note that the performance can only be compared indirectly as it uses a different set of LoRA adapters and training tasks. Furthermore, there are likely differences in the benchmark evaluation prompts.

In most experiments, we use `Mistral-7B-Instruct` (Jiang et al., 2023) as the base LLM model except in Tables 3 and 4 where `Llama-3.1-8B-Instruct` and `Gemma-2-2b-Instruct` are used as the base models, respectively. We use `gte-large-en-v1.5` (Li et al., 2023; Zhang et al., 2024) for extracting the task embedding from a natural language task description. All LoRA adapters are of rank 8 and only target the query and the value projection modules in every attention block of the base LLM (totaling 3.4M parameters). With this LoRA configuration, **L** , **M** , and **S** have 55M, 34M, and 5M trainable parameters respectively. We utilize the SNI dataset (Wang et al., 2022) for training LoRA adapters. We use a subset of 500 tasks following Brüel-Gabrielsson et al. (2024), 10 of which are manually chosen while the rest are randomly sampled. We use 11 tasks for hold-out validation, and remove 10 datasets due to data contamination from the evaluation benchmark tasks, leaving 479 datasets for training. All samples are in English.

For evaluation, we choose 10 widely used benchmarks that collectively cover a variety of LLM capability assessments, e.g., reasoning, math, science, coding, and world knowledge. Specifically, we include the following benchmarks: Arc-challenge (ArcC) and Arc-easy (ArcE) (Clark et al., 2018), BoolQ (Clark et al., 2019), GSM8K (Cobbe et al., 2021), Hellaswag (HS) (Zellers et al., 2019), OpenBookQA (OQA) (Mihaylov et al., 2018), PIQA (Bisk et al., 2020), Winogrande (WG) (Keisuke et al., 2019), HumanEval (HE) (Chen et al., 2021), and MBPP (Austin et al., 2021). Task descriptions for the training datasets and the benchmarks are fully generated, as described in Appendix G. When we use a language task embedding as a part of the input, we average `T2L` performance using three descriptions for each benchmark.

## 4.1 LoRA COMPRESSION

Table 1: Benchmark performance of `T2L` trained via reconstruction loss on 9 benchmark tasks. **Green highlight** indicates that `T2L` outperforms the benchmark-specific LoRA adapters.

| | ArcC (acc) | ArcE (acc) | BQ (acc) | GSM8K (acc) | HS (acc) | OQA (acc) | PIQA (acc) | WG (acc) | MBPP (pass@1) | Avg. (9 tasks) |
|---|---|---|---|---|---|---|---|---|---|---|
| Base model | 65.4 | 77.8 | 71.6 | 40.9 | 49.7 | 54.2 | 72.8 | 45.0 | 43.1 | 55.8 |
| **One-Hot Task Embeddings** | | | | | | | | | | |
| `T2L` (Recon) **L** | 76.4 | 89.9 | 89.4 | 53.8 | 92.6 | 85.0 | 69.7 | 51.2 | 52.6 | 73.4 |
| `T2L` (Recon) **M** | 76.7 | 89.9 | 89.4 | 53.2 | 92.6 | 85.0 | 69.9 | 51.4 | 52.9 | 73.4 |
| `T2L` (Recon) **S** | 75.2 | 88.8 | 87.4 | 50.9 | 89.1 | 75.6 | 83.9 | 58.1 | 48.1 | 73.0 |
| **Task Description Embeddings** | | | | | | | | | | |
| `T2L` (Recon) **L** | 76.6 | 89.8 | 89.4 | 53.9 | 92.6 | 85.0 | 69.6 | 51.2 | 51.8 | 73.3 |
| `T2L` (Recon) **M** | 76.5 | 89.9 | 89.4 | 53.9 | 92.5 | 84.9 | 70.4 | 51.6 | 52.8 | 73.5 |
| `T2L` (Recon) **S** | 75.4 | 88.8 | 87.8 | 49.1 | 89.7 | 76.7 | 84.2 | 56.9 | 48.0 | 73.0 |
| Task-specific LoRAs | 76.6 | 89.9 | 89.4 | 53.5 | 92.6 | 85.0 | 69.9 | 51.1 | 52.1 | 73.3 |

In this experiment, we aim to investigate whether `T2L` can recover the performance of trained LoRAs via reconstruction training. For quality control and consistent evaluation, we train a task-specific LoRA (oracle) on the training split of each benchmark task, collectively forming a library of LoRAs. Table 1 shows the benchmark performance of `T2L` trained by distilling 9 benchmark-specific LoRAs using either one-hot or natural language task embeddings from `gte-large-en-v1.5`. We note that the

benchmark tasks are indirectly seen during training by **T2L** as it learns to distill benchmark-specific LoRAs. We can see that **T2L** fully recovers the performance of the oracle adapters with both task embedding types. Notably, **T2L** outperforms task-specific LoRAs on several benchmarks (highlighted in green). We hypothesize that the gain comes from the lossy compression of the target LoRAs, which acts as a regularization on the already trained LoRA weights. This effect is most apparent on PIQA and WG benchmarks, where the oracle LoRA overfits and performs worse than the base model.

Next, we explore whether **T2L** conditioned on one-hot task vectors can maintain the oracle single-task LoRA's performance when using an increasing number of training tasks. Fig. 3 shows the performance of one-hot **T2L** on the test splits of a subset of 10 SNI training tasks with varying degrees of final average training L1 reconstruction error. We train various **T2L** instances for each architecture using $\{16, 32, 64, 128, 256, 489\}$ training tasks, effectively increasing the training reconstruction error. Although **T2L** fully recovers the oracles' performance when the reconstruction loss is less than $10^{-4}$, the performance drops as the training error increases. This result suggests that **T2L** learns a lossy compression of the target LoRAs. Still, we find that all **T2L** architectures can maintain around $65\%$ of oracles' performance, and the performance does not drop further even at $> 8 \times 10^{-4}$ per element L1 error. Despite the performance drop, we show that increasing the number of training tasks is beneficial, increasing generalization of **T2L** when generating LoRAs for unseen tasks in Section 5.1.

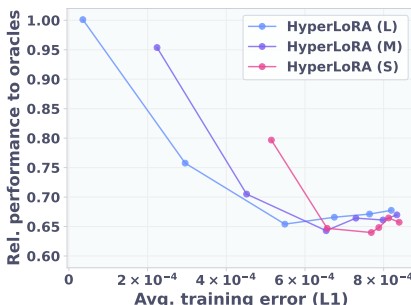

Figure 3: Relative performance (y-axis) and training reconstruction error (x-axis) of **T2L** instances trained with an increasing number of tasks ($\{16, 32, 64, 128, 256, 489\}$ from left to right of each line).

## 4.2 ZERO-SHOT LORA GENERATION

Table 2: Zero-shot performance on unseen benchmark tasks. SFT-trained **T2L** generates LoRAs based on unseen task descriptions. Its performance is an average of three generated LoRAs, each with a different instance of task descriptions. Arrow Routing results are taken from Ostapenko et al. (2024). **Green highlight** indicates high performance than that of the benchmark-specific LoRA adapters. **Bold numbers** are used when the performance is higher than the multi-task LoRA.

| | ArcC (acc) | ArcE (acc) | BQ (acc) | HS (acc) | OQA (acc) | PIQA (acc) | WG (acc) | MBPP (pass@1) | Avg. (8 tasks) | GSM8K (acc) | HE (pass@1) | Avg. |
|---|---|---|---|---|---|---|---|---|---|---|---|---|
| **No Test-Time Adaptation** | | | | | | | | | | | | |
| Mistral-7B-Instruct | 65.4 | 77.8 | 71.6 | 49.7 | 54.2 | 72.8 | 45.0 | 43.1 | 60.0 | 40.9 | 37.2 | 55.8 |
| Prepending task desc. | 72.0 | 85.8 | 67.6 | 58.9 | 63.4 | 77.9 | 59.0 | 41.6 | 65.8 | 40.9 | 39.0 | 60.6 |
| 3-shot ICL | 72.1 | 85.9 | 71.7 | 59.0 | 66.2 | 76.2 | 58.0 | 42.6 | 66.5 | 40.9 | 37.2 | 61.0 |
| Average LoRA | 70.7 | 84.4 | 75.4 | 59.9 | 59.0 | 78.0 | 54.3 | 47.1 | 66.1 | 42.4 | 37.8 | 60.9 |
| Multi-task LoRA | 76.2 | 88.3 | 85.5 | 65.2 | 68.0 | 81.8 | 62.4 | 48.1 | 71.9 | 47.5 | 39.6 | 66.3 |
| **Zero-Shot Adaptation** | | | | | | | | | | | | |
| Arrow Routing | 60.9 | 86.2 | **87.6** | **80.8** | 48.6 | **83.0** | **68.5** | **50.2** | 70.7 | N/A | 28.7 | N/A |
| Hyperdecoders (per-instance) | **76.6** | **88.5** | 83.9 | 65.2 | **76.6** | **81.3** | **64.9** | **51.6** | **73.6** | 43.6 | **40.9** | **67.3** |
| T2L (SFT) S | 76.0 | **88.7** | 83.8 | **68.0** | 71.6 | **82.3** | **61.0** | 41.2 | 71.6 | 47.3 | 39.0 | 65.9 |
| T2L (SFT) M | **77.2** | **89.0** | 84.3 | 65.1 | **76.1** | 81.8 | **64.0** | **50.5** | **73.5** | 45.2 | **41.3** | 67.5 |
| T2L (SFT) L | **77.5** | **88.9** | 85.0 | 66.5 | **75.5** | **82.1** | **64.2** | **51.9** | **73.9** | 45.8 | 39.2 | 67.7 |
| **Oracle** | | | | | | | | | | | | |
| Task-specific LoRAs | 76.6 | 89.9 | 89.4 | 92.6 | 85.0 | 69.9 | 51.1 | 52.1 | 75.8 | 53.5 | N/A | N/A |

Here, we explore whether **T2L** can generate useful LoRA adapters for unseen tasks. We train **T2L** with SFT on 479 SNI tasks, each with 128 task descriptions. For each data point in a training minibatch, we sample a description from the corresponding dataset in an online fashion. Table 2 shows the zero-shot performance on 10 benchmark tasks. Here, we present the best model of each variant from our scaling experiment in Section 5.2. We observe that a multi-task LoRA adapter performs well on the benchmarks despite no additional fine-tuning. Still, there is a performance gap between task-specific LoRAs and MT LoRA. We observe that SFT-trained **T2L** indeed generates useful LoRAs, thus improving over the multi-task LoRA adapter consistently and across benchmarks (indicated by bold numbers). Notably, even though **T2L** cannot fully bridge the performance gap with task-specific LoRAs, it outperforms the oracles on a subset of tasks (highlighted in green).

### 4.3 GENERALIZATION TO Llama AND Gemma MODELS

Table 3: Zero-shot performance with `Llama-3.1-8B-Instruct` as the base language model.

| | ArcC (acc) | ArcE (acc) | BQ (acc) | GSM8K (acc) | HS (acc) | OQA (acc) | PIQA (acc) | WG (acc) | HE (pass@1) | MBPP (pass@1) | Avg. |
|---|---|---|---|---|---|---|---|---|---|---|---|
| Llama-3.1-8B-Instruct | 73.3 | 90.6 | 80.4 | 75.7 | 66.6 | 75.4 | 79.8 | 55.3 | 66.5 | 68.7 | 73.2 |
| 3-shot ICL | 80.7 | 91.9 | 80.0 | 75.7 | 59.3 | 77.6 | 80.9 | 61.3 | 66.5 | 70.4 | 74.4 |
| Prepending task desc. | 80.2 | 92.5 | 79.9 | 75.7 | 69.8 | 78.4 | 81.7 | 62.4 | 68.3 | 70.2 | 75.9 |
| Multi-task LoRA | 82.0 | 92.8 | 83.3 | 77.6 | 70.8 | **81.8** | **83.8** | **60.3** | 63.4 | 69.4 | 76.5 |
| T2L (SFT) L | **82.4** | **92.9** | **84.4** | **79.1** | **72.8** | 81.8 | 81.2 | 60.0 | **64.6** | **69.9** | **76.9** |

Table 4: Zero-shot performance with `Gemma-2-2B-Instruct` as the base language model.

| | ArcC (acc) | ArcE (acc) | BQ (acc) | GSM8K (acc) | HS (acc) | OQA (acc) | PIQA (acc) | WG (acc) | HE (pass@1) | MBPP (pass@1) | Avg. |
|---|---|---|---|---|---|---|---|---|---|---|---|
| Gemma-2-2B-Instruct | 73.7 | **89.9** | 81.0 | 55.6 | 55.2 | 71.0 | 71.0 | 53.8 | **43.9** | 12.3 | 60.7 |
| 3-shot ICL | 72.4 | 88.9 | **82.5** | 55.6 | 55.7 | 72.6 | 67.6 | 53.7 | **43.9** | 43.1 | 63.6 |
| Prepending task desc. w/ ICL | 72.4 | 88.9 | **82.5** | 55.6 | 55.7 | 72.6 | 67.6 | 53.7 | **43.9** | 43.1 | 63.6 |
| Multi-task LoRA w/ ICL | 73.5 | 89.4 | 81.6 | **57.2** | 59.5 | **74.6** | 69.4 | 58.1 | 39.0 | 50.4 | 65.2 |
| T2L (SFT) L w/ ICL | **74.0** | 89.8 | 81.8 | 55.1 | **62.5** | 73.9 | **75.2** | **58.7** | 41.5 | **51.5** | **66.4** |

In this section, we explore the generality of our proposed architectures to different model families and sizes. Tables 3 and 4 show the benchmark performance of T2L L compared to various baselines using `Llama-3.1-8B-Instruct` and `Gemma-2-2B-Instruct` as the base models, respectively. With `Gemma` base model, we utilize ICL for all approaches as it drastically improves the performance on the MBPP benchmark. We see that T2L consistently outperforms the baselines across all tested models with varying model sizes and architectures. We note that T2L are trained with the same set of hyperparameters across base models.

## 5 ABLATIONS

### 5.1 SCALING THE NUMBER OF TRAINING TASKS WITH FIXED COMPUTE

Table 5: Benchmark performance of SFT-trained T2L with varying numbers of training tasks. We show results with $\{64, 128, 256, 479\}$ tasks. ▲ (▼) indicates increased (decreased) performance compared to the previous increment in the number of training tasks and training budget.

| | Number of tasks | ArcC (acc) | ArcE (acc) | BQ (acc) | GSM8K (acc) | HS (acc) | OQA (acc) | PIQA (acc) | WG (acc) | HE (pass@1) | MBPP (pass@1) | Avg. |
|---|---|---|---|---|---|---|---|---|---|---|---|---|
| T2L (SFT) L | 479 | 77.2 | 89.0 | 85.0 | 46.3 | 66.5 | 73.6 | 82.6 | 61.8 | 39.2 | 44.3 | 66.6 ▼ |
| | 256 | 76.6 | 89.1 | 84.8 | 47.0 | 67.7 | 73.5 | 82.8 | 62.4 | 39.6 | 51.0 | 67.5 ▲ |
| | 128 | 76.2 | 89.0 | 85.3 | 46.2 | 67.9 | 71.7 | 82.6 | 59.9 | 40.5 | 51.3 | 67.0 ▲ |
| | 64 | 75.5 | 88.0 | 84.5 | 43.9 | 65.5 | 70.7 | 80.5 | 59.5 | 39.8 | 51.7 | 66.0 |
| T2L (SFT) M | 479 | 77.5 | 89.0 | 85.0 | 45.8 | 66.5 | 71.9 | 82.1 | 61.4 | 41.3 | 50.1 | 67.1 ▲ |
| | 256 | 76.1 | 89.2 | 85.3 | 45.4 | 65.6 | 72.7 | 81.7 | 62.3 | 36.8 | 50.6 | 66.5 ▲ |
| | 128 | 75.5 | 87.8 | 85.3 | 46.1 | 66.6 | 71.6 | 81.7 | 62.2 | 39.8 | 44.9 | 66.1 ▲ |
| | 64 | 73.6 | 87.7 | 84.5 | 43.2 | 64.6 | 70.5 | 79.9 | 56.0 | 40.7 | 51.4 | 65.2 |
| T2L (SFT) S | 479 | 75.8 | 88.5 | 83.9 | 45.6 | 64.2 | 71.9 | 82.3 | 61.5 | 36.2 | 45.0 | 65.5 ▲ |
| | 256 | 76.1 | 88.4 | 83.0 | 47.3 | 65.0 | 71.7 | 82.5 | 58.1 | 36.2 | 39.1 | 64.8 ▲ |
| | 128 | 75.6 | 87.7 | 84.9 | 46.5 | 65.7 | 72.7 | 81.0 | 59.6 | 39.0 | 28.1 | 64.1 ▼ |
| | 64 | 75.4 | 88.4 | 85.0 | 43.1 | 64.8 | 70.7 | 81.5 | 51.6 | 39.4 | 46.7 | 64.7 |

We study the impact of the number of training tasks on the zero-shot benchmark performance of T2L in the SFT setting, where all T2L instances are trained for roughly the same number of gradient steps (see details in Appendix D). Overall, we find that increasing the number of training tasks improves the average zero-shot benchmark performance of the hypernetwork (Fig. 1 and Table 5). This result hints at the plausible scalability of T2L and positive transfer between tasks.

Table 6: Benchmark performance of SFT-trained T2L with varying numbers of training tasks.

| | Number of tasks | Max SGD steps | ArcC (acc) | ArcE (acc) | BQ (acc) | GSM8K (acc) | HS (acc) | OQA (acc) | PIQA (acc) | WG (acc) | HE (pass@1) | MBPP (pass@1) | Avg. |
|---|---|---|---|---|---|---|---|---|---|---|---|---|---|
| T2L (SFT) L | 479 | 1M | 77.5 | 88.9 | 85.0 | 45.8 | 66.5 | 75.5 | 82.1 | 64.2 | 39.2 | 51.9 | 67.7 ▲ |
| | 256 | 640K | 77.3 | 88.1 | 84.3 | 46.0 | 64.5 | 75.7 | 81.9 | 64.0 | 39.8 | 52.1 | 67.4 ▲ |
| | 128 | 320K | 76.6 | 88.4 | 85.2 | 46.1 | 67.0 | 74.3 | 81.6 | 55.0 | 38.2 | 45.7 | 65.8 ▼ |
| | 64 | 160K | 75.5 | 88.0 | 84.5 | 43.9 | 65.5 | 70.7 | 80.5 | 59.5 | 39.8 | 51.7 | 66.0 |
| T2L (SFT) M | 479 | 1M | 77.2 | 89.0 | 84.3 | 45.2 | 65.1 | 76.1 | 81.8 | 64.0 | 41.3 | 50.5 | 67.5 ▲ |
| | 256 | 640K | 75.9 | 89.3 | 85.0 | 47.0 | 65.3 | 73.7 | 81.6 | 63.2 | 39.8 | 48.6 | 66.9 ▲ |
| | 128 | 320K | 74.9 | 88.3 | 85.5 | 44.9 | 64.8 | 72.8 | 80.7 | 61.6 | 42.9 | 43.5 | 66.0 ▲ |
| | 64 | 160K | 73.6 | 87.7 | 84.5 | 43.2 | 64.6 | 70.5 | 79.9 | 56.0 | 40.7 | 51.4 | 65.2 |
| T2L (SFT) S | 479 | 1M | 77.7 | 88.3 | 85.0 | 46.3 | 65.3 | 73.9 | 82.4 | 61.9 | 34.6 | 36.6 | 65.2 ▼ |
| | 256 | 640K | 76.0 | 88.7 | 83.8 | 47.3 | 68.0 | 71.6 | 82.3 | 61.0 | 39.0 | 41.2 | 65.9 ▲ |
| | 128 | 320K | 74.9 | 88.0 | 84.5 | 44.4 | 66.2 | 72.2 | 82.0 | 59.3 | 39.0 | 47.3 | 65.8 ▲ |
| | 64 | 160K | 75.4 | 88.4 | 85.0 | 43.1 | 64.8 | 70.7 | 81.5 | 51.6 | 39.4 | 46.7 | 64.7 |

## 5.2 INCREASING TRAINING COMPUTE PROPORTIONAL TO THE NUMBER OF TRAINING TASKS

As the performance of the L variant drops after increasing the number of training tasks from 256 to 479 with a fixed training budget (Table 5), we investigate whether increasing the training budget would allow T2L to scale more gracefully. Specifically, we increase the training budget proportionally to the dataset size on all variants. Table 6 shows that, after increasing the training budget, L benefits from the additional training tasks. Additionally, M improves over training runs with a fixed budget when using 256 or more training tasks. However, S does not benefit from extended training with 479 tasks, potentially due to its limited model capacity.

## 5.3 TRAINING DESCRIPTION SOURCES

Table 7: Benchmark performance of SFT-trained T2L with two different training description sources.

| | ArcC (acc) | ArcE (acc) | BQ (acc) | GSM8K (acc) | HS (acc) | OQA (acc) | PIQA (acc) | WG (acc) | HE (pass@1) | MBPP (pass@1) | Avg. |
|---|---|---|---|---|---|---|---|---|---|---|---|
| T2L (SFT) L | 77.5 | 88.9 | 85.0 | 45.8 | 66.5 | 75.5 | 82.1 | 64.2 | 39.2 | 51.9 | 67.7 |
| T2L (SFT) L w/ SNI def. | 75.3 | 87.4 | 85.0 | 45.9 | 63.6 | 73.5 | 80.9 | 61.8 | 38.2 | 53.8 | 66.5 |

In this experiment, we explore the impact of the sources of the training task descriptions: SNI and chatGPT (Appendix G) Table 7 shows that using task definitions provided by the SNI datasets reduces the zero-shot benchmark performance of T2L. As the SNI datasets are crowd-sourced, we hypothesized that the task descriptions might have inconsistent template or varied levels of details. Thus, it is harder for T2L to learn and generalize.

## 5.4 TASK EMBEDDING MODELS

Table 8 shows the zero-shot benchmark performance with two different embedding models: `gte-large-en-v1.5` and `Mistral-7B-Instruct`. For the `gte` model, we extract a task description by presenting the activation of the `CLS` token in the last layer (1024D) as the model is a bidirectional model.

Table 8: Zero-shot benchmark performance of T2L trained via SFT on 128 tasks.

| | gte | | | Mistral | | |
|---|---|---|---|---|---|---|
| | S | M | L | S | M | L |
| Avg. Benchmark performance | 65.8 | 66.0 | 65.8 | 64.7 | 66.2 | 66.0 |
| **Avg.** | | **65.9** | | | **65.6** | |

For `Mistral`, we use the activation of the last token in the sequence (4096D) to represent a given description (BehnamGhader et al., 2024). Table 8 shows the results with the two embedding models used for T2L SFT training on 128 tasks. Both embedding models yield T2L instances with comparable generalization capability (65.9 vs 65.6 on average), suggesting T2L robustness to specific text embedding methods.

## 5.5 VARYING TASK DESCRIPTIONS

We investigate the impact of input task descriptions on the performance of generated LoRAs. We use four types of task descriptions:

- **Train:** Training descriptions
- **Eval:** Unseen descriptions
- **Random strings:** Random literal strings
- **Train (random):** Training descriptions randomly sampled from other benchmarks

Table 9: Benchmark performance of **T2L** trained via reconstruction on 9 benchmark tasks.

| | | Aligned | | Unaligned | |
|---|---|---|---|---|---|
| | | **Train** | **Eval** | **Train (random)** | **Random strings** |
| T2L | L | 73.3 | 73.6 | 49.1 | 68.2 |
| T2L | M | 73.5 | 70.2 | 49.5 | 68.5 |
| T2L | S | 73.0 | 72.9 | 55.7 | 53.9 |
| **Avg.** | | **73.3** | **72.2** | **51.4** | **63.5** |

For each description type, we use the `gte-large-en-v1.5` embedding and report the average performance using three descriptions. The four types can be grouped into two categories based on the alignment between the descriptions and the tasks: aligned (**Train**, **Eval**) and unaligned (**Train (random)** and **Random strings**). Note that we use reconstruction-trained **T2L** in this experiment. That is, the hypernetwork has seen training descriptions of the benchmarks during training. We observe a performance gap between the two description categories. Specifically, training and evaluation descriptions generate the best performing LoRAs, matching the performance of oracle LoRAs, despite the evaluation descriptions being unseen. These results suggest that **T2L** are robust to changes in the task description as long as the descriptions are aligned with the task. On the other hand, if the descriptions are not aligned with the task at hand, the generated LoRAs will not perform as well, as indicated by the performance of the unaligned group.

## 5.6 TRAINING SCHEMES

In this section, we investigate the zero-shot performance of SFT-trained and reconstruction-trained **T2L**. All model instances are trained with roughly equal wall-clock time of 10 hours (see Appendix D for details). From Table 10, we can see a clear performance gap between reconstruction and SFT training schemes. Specifically,

Table 10: Zero-shot benchmark performance of **T2L** trained via reconstruction and SFT.

| | Recon | | | SFT | | |
|---|---|---|---|---|---|---|
| | **S** | **M** | **L** | **S** | **M** | **L** |
| Benchmark performance | 61.8 | 61.7 | 62.0 | 64.8 | 66.5 | 67.5 |
| **Avg.** | | **61.8** | | | **66.3** | |

SFT produces **T2L** instances that perform significantly better than that of reconstruction training (66.3 vs 61.83 benchmark performance averaged over model architectures). We attribute the performance difference to the library of LoRAs needed for reconstruction training. For reconstruction-trained **T2L** to generalize, the target LoRA adapters of similar tasks should be clustered in some latent manifold. In contrast, SFT training does not need pre-trained task-specific LoRA adapters, thus sidestepping this challenge via end-to-end learning. In Section 6.1, we show that pre-trained adapters for similar tasks do not live nearby in the weight space, supporting our claim of a potential problem when reconstructing pre-trained LoRA adapters.

## 6 ANALYSIS

### 6.1 LORAS OF SIMILAR TASKS

Here, we investigate the relationship between LoRA adapters by inspecting their similarity in the parameter space, performance on the benchmarks, and similarity of their description embeddings. To measure adapter similarity, we compute the cosine similarity of the concatenation of flattened low-rank $A$ and $B$ matrices of all layers. In the top row of Fig. 4, we plot the adapters' similarity against task description similarity (using the mean embedding of each task). We find no correlation between the cosine similarity of the adapters' weights (y-axis) and the task embedding similarity (x-axis) indicated by near-zero Pearson correlation coefficients.

In the bottom row of Fig. 4, we change the y-axis from adapters' relative benchmark performance to benchmark-specific adapters. We find a positive correlation between the relative benchmark performance of SNI-trained adapters and the task embedding similarity. That is, adapters perform better on a benchmark if their task descriptions are similar to those of the benchmark. However, despite their similar functionalities, adapters with similar descriptions are not similar in the parameter space. We believe that this relationship has a significant impact on the limited generalization of reconstruction-trained **T2L**. We further discuss this topic in Appendix F.

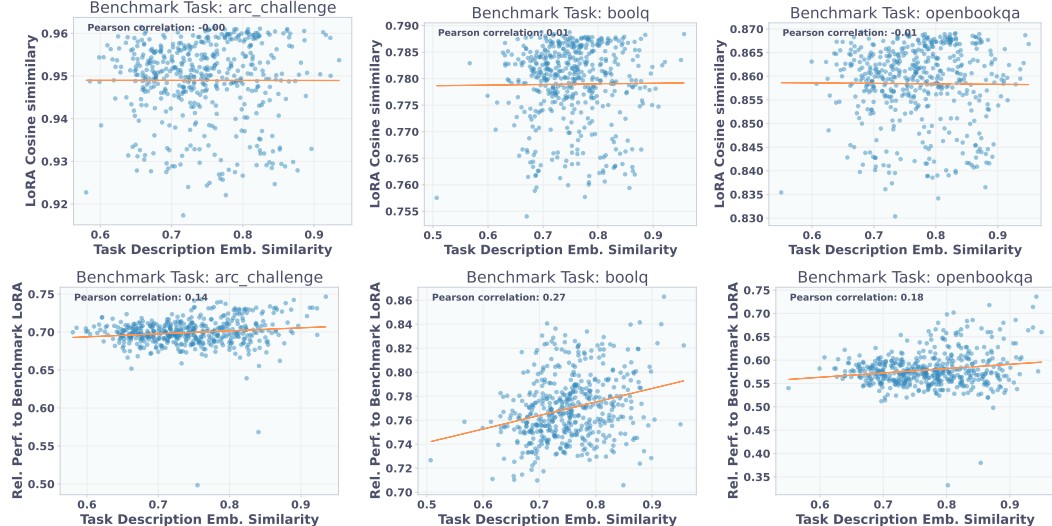

Figure 4: **Top row:** Each plot shows the similarity between a benchmark LoRA adapter and 489 SNI-trained adapters in the weight space (y-axis) against their similarity in the task embedding space (x-axis). **Bottom row:** Each plot shows SNI-trained adapters' performance relative to a benchmark adapter (y-axis) with the same x-axis. We can see that LoRAs with similar description embeddings to the benchmarks' perform better in those benchmarks, suggesting their shared functionalities. However, LoRAs with similar functionalities are not nearby in the parameter space.

## 6.2 VISUALIZATION OF **T2L** ACTIVATIONS

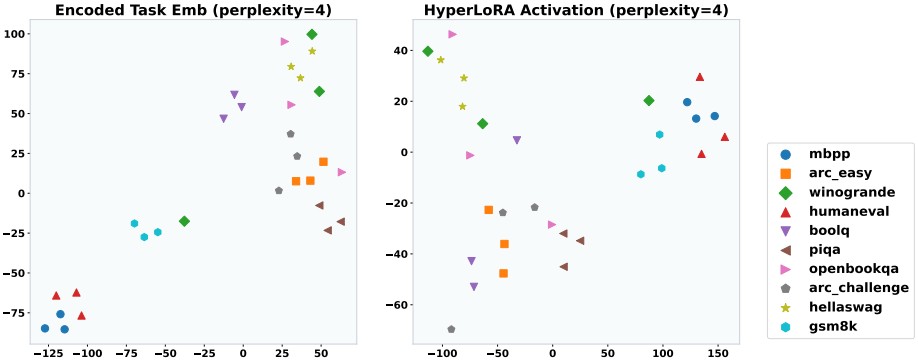

Figure 5: 2D t-SNE projection of activations of **T2L**'s task encoder (left) and activations of the last MLP block (right) grouped by benchmark tasks (represented by colors). We probe **T2L** with unseen three task descriptions per benchmark. We can see activations clustering in both plots, indicating that **T2L** indeed learns to generate LoRAs tailored to specific tasks.

Next, we aim to understand **T2L** further and see whether it generates task-specific LoRA adapters for unseen tasks with unseen descriptions. We probe our best-performing model in the zero-shot evaluation setting, SFT **T2L** **M** trained on 256 training tasks. We probe the model on all the benchmark tasks, each with three unseen descriptions. Fig. 5 shows the 2D t-SNE projection of **T2L**'s task encoder activation and the output of the last MLP block. We can see a clear clustering in both projection plots based on the tasks (colors and shapes). **T2L** generates different adapters for different tasks, confirming that **T2L** indeed performs task-specific adaptation 'on the fly'. Moreover, similar tasks, e.g., MBPP and HumanEval, are clustered together in both plots, suggesting that SFT-trained **T2L** produces similar adapters for semantically similar tasks.

## 7 RELATED WORK

**Hypernetworks for Adaptation**: Hypernetworks (Ha et al., 2016) provide a general indirect encoding method for neural network weights. They have been applied to different architectures (e.g., in attention, Schug et al., 2024) and training paradigms (e.g., in continual learning, Von Oswald et al., 2019). Here, we focus on generating low-rank adapters using natural language instructions. Previous work (Mahabadi et al., 2021; He et al., 2022; Ortiz-Barajas et al., 2024) considers hypernetworks for LLM adaptation in a multi-task context but only uses learned task identifiers instead of natural language for adaptation. Thus, these approaches do not enable task-wise zero-shot generalization.

**Hypernetworks for Zero-Shot LLM Adaptation:** Xiao et al. (2023) explore the use of HyperLoRA on a limited set of English dialects; they only consider five dialects, one of which is unseen. Furthermore, the hypernetwork relies on an expert-based transformation of the dialects, limiting the possibility of generalization. Mu et al. (2024) propose Gisting, a method that learns to compress an in-context task description to prefix tokens, allowing the language model to follow instructions with less token count. However, Gisting is limited to prefix tokens—only influencing the attention matrices of the base model. Thus, prefix tokens are less flexible compared to LoRA that can modify different parts of LLMs, e.g., MLP blocks. Hyperdecoders (Ivison & Peters, 2022) is a hypernetwork that generates adapters on the fly based on the input sequence. While per-sequence adaptation is desirable for benchmark evaluation—where the LLM should always output the correct answer—we argue that description-based adaptation gives more control to users since they can steer the LLM in creative ways based on user-generated descriptions. Furthermore, the generated adapters cannot be efficiently fused into the base model, leading to significant overhead for each query.

Closely related to our work are HyperTuning (Phang et al., 2023), HNET-LM (Deb et al., 2022), and HINT (Ivison et al., 2023). Differ from prior work that heavily focuses on pre-trained encoder-decoder models, e.g., T5 (Raffel et al., 2020) or BART (Lewis, 2019), we use frontier instruction fine-tuned models as the base models, i.e., `Mistral`, `Llama`, `Gemma`. Further, prior work typically relies on initializing a part of their hypernetworks from the base model (e.g., tying task encoder's weights to the base model) to achieve good performance or stable training as opposed to ours that are task-embedder agnostic and can freely change the task embedding model (Section 5.4). Additionally, our work utilizes generated descriptions instead of the ones provided by the SNI dataset. Overall, our work improves upon prior work in several ways, including achieving task-wise zero-shot generalization on various frontier instruction-tuned language models, simpler and more general hypernetwork input requirements, investigation of training regimes, and more comprehensive experiments, ablations, and analyses.

Concurrent to our work, Lv et al. (2024) propose a similar approach that utilizes a hypernetwork to generate LoRA adapter at inference time. However, their hypernetwork assumes that the context vector provided to the hypernetwork contains few-shot examples. In contrast, **T2L** only assumes a task description, which users can produce themselves within seconds.

## 8 CONCLUSION

**Discussion**. We rely on generated descriptions from `GPT-4o mini` to ensure high-quality and consistent task descriptions. It is plausible that when **T2L** is deployed in real-world scenarios, users might not input high-quality descriptions, which could cause performance degradation on generated adapters. Our results have primarily focused on LLM adaptation. However, **T2L** can be directly applied to other LLMs or adapting vision language models. Finally, the potential for **T2L** trained on a smaller base model to transfer effectively to larger models within the same architecture class remains an open area for exploration.

**Limitations**. We only consider LoRA as the output space of the hypernetwork. We believe there are more efficient ways to modulate LLMs given a text description, e.g., directly modulate the activations of the base model. Additionally, we believe the compression achieved by **T2L** can be further optimized using well-designed inductive biases. Though **T2L** performs better when trained on more tasks, it is yet to take full advantage of scaling up to a larger set of training tasks shown in Section 5.1. Finally, although **T2L** exhibits robustness and signs of scalability, it still does not reach the benchmark performance of task-specific LoRAs in a zero-shot manner. Achieving such potent zero-shot adaption is still one of the biggest challenges for **T2L**.

**Future Work**. We plan to expand `T2L` across multiple architectures and explore the potential for transfer between different model sizes. In addition, we intend to rigorously study the scaling laws of LoRA compression as the number of `T2L` parameters, dataset, and compute increases. Finally, our goal is to provide an openly accessible service for various `T2L` configurations. We envision a user-friendly platform where individuals could generate and download fine-tuned adapters by simply prompting a model with a minimal chat interface.

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

## A HYPERPARAMETER SETTINGS

Table 11: Hyperparameters for training a task-specific LoRA adapter.

| Hyperparameters | Task-specific LoRA | HyperLoRA (SFT) | HyperLoRA (recon) |
|---|---|---|---|
| Batch size | 8 | 8 | Number of the target LoRAs |
| Gradient accumulation steps | 1 | 1 | 1 |
| Max learning rate | $8 \times 10^{-5}$ | $2.5 \times 10^{-5}$ | $10^{-3}$ |
| Max gradient norm | 1.0 | 1.0 | 1.0 |
| NEFTune noise alpha | 5.0 | 5.0 | 5.0 |
| Warmup fraction | 0.1 | 0.1 | 0.1 |
| Learning rate scheduler | Linear with warm up | Linear with warm up | Linear with warm up |

```
{
  "alpha_pattern": {},
  "auto_mapping": null,
  "base_model_name_or_path": "models/Mistral-7B-Instruct-v0.2",
  "bias": "none",
  "fan_in_fan_out": false,
  "inference_mode": true,
  "init_lora_weights": true,
  "layer_replication": null,
  "layers_pattern": null,
  "layers_to_transform": null,
  "loftq_config": {},
  "lora_alpha": 16,
  "lora_dropout": 0.05,
  "megatron_config": null,
  "megatron_core": "megatron.core",
  "modules_to_save": null,
  "peft_type": "LORA",
  "r": 8,
  "rank_pattern": {},
  "revision": null,
  "target_modules": [
    "q_proj",
    "v_proj"
  ],
  "task_type": "CAUSAL_LM",
  "use_dora": false,
  "use_rslora": true
}
```

Listing 1: The parameter-efficient fine-tuning (PEFT) config for all LoRA adapters.

Table 11 and Listing 1 show the training configuration of all models trained in this work. For LoRA reconstruction training, each prediction target is an entirety of a LoRA adapter. That is, there is a total of 489 training samples for 489 SNI tasks. Thus, we increase the epochs to $100,000$ to ensure that **T2L** converges.

## B ADDITIONAL DETAILS OF HYPERLORA ARCHITECTURES

Listings 2 and 3 shows the details of the backbone of **T2L**. Specifically, the size of the module and layer embedding ($E[m]$ and $E[l]$) is 32D. Together, they form a dictionary of 34 learnable embeddings (32 layers + 2 target modules). The task encoder is a linear layer that takes in a text embedding (1024D for the `gte` embedding and 4096D for `Mistral` embedding) and outputs a 64D vector. The

```
Hypermod: HyperModulator(
  (task_encoder): TaskEncoder(
    (mlp): Sequential(
      (0): Linear(in_features=1024, out_features=64, bias=True)
      (1): LayerNorm((64,), eps=1e-05, elementwise_affine=True)
    )
  )
  (layer_depth_encoder): Sequential(
    (0): Embedding(32, 32)
    (1): LayerNorm((32,), eps=1e-05, elementwise_affine=True)
  )
  (layer_type_encoder): Sequential(
    (0): Embedding(2, 32)
    (1): LayerNorm((32,), eps=1e-05, elementwise_affine=True)
  )
  (mixer): Sequential(
    (0): Linear(in_features=128, out_features=512, bias=True)
    (1): SiLU()
    (2): Dropout(p=0.05, inplace=False)
    (3): Linear(in_features=512, out_features=128, bias=True)
    (4): SiLU()
    (5): Dropout(p=0.05, inplace=False)
  )
  (mlp1): MLPResidualBlock(
    (mlp): Sequential(
      (0): LayerNorm((128,), eps=1e-05, elementwise_affine=True)
      (1): Linear(in_features=128, out_features=512, bias=True)
      (2): SiLU()
      (3): Dropout(p=0.05, inplace=False)
      (4): Linear(in_features=512, out_features=128, bias=True)
      (5): SiLU()
      (6): Dropout(p=0.05, inplace=False)
    )
  )
  (mlp2): MLPResidualBlock(
    (mlp): Sequential(
      (0): LayerNorm((128,), eps=1e-05, elementwise_affine=True)
      (1): Linear(in_features=128, out_features=512, bias=True)
      (2): SiLU()
      (3): Dropout(p=0.05, inplace=False)
      (4): Linear(in_features=512, out_features=128, bias=True)
      (5): SiLU()
      (6): Dropout(p=0.05, inplace=False)
    )
  )
  (mlp3): Sequential(
    (0): LayerNorm((128,), eps=1e-05, elementwise_affine=True)
    (1): Linear(in_features=128, out_features=512, bias=True)
    (2): SiLU()
    (3): Dropout(p=0.05, inplace=False)
    (4): Linear(in_features=512, out_features=512, bias=True)
    (5): SiLU()
  )
)
```

Listing 2: Detailed backbone architecture.

encoded task, module, and layer embedding are concatenated and then fed into mlp0 followed by a residual MLP block mlp1. At this point, for **M** and **S**, we add a 128D A/B embbedding to the residual stream. The output is then fed to another residual MLP block mlp2. At this point, for **S**, we add a 128D rank embedding to the residual stream. After this, we feed the activation to the last MLP block. The output of the last MLP block is then fed to a linear head, whose output size is as follows:

- **L** : $2 \times r \times d$ giving both $A$ and $B$ matrices

- **M** : $r \times d$ giving a low-rank matrix $A$ or $B$ depending on the A/B embedding

- **S** : $d$ giving a rank of a low-rank matrix depending on both the A/B embedding and the rank embedding.

For ease of explanation, we assume $d$ is the same for the input and the output space of a linear transformation. In practice, $d_{\text{in}} = d_{\text{out}} = 4096$ for q_proj module and $d_{\text{in}} = 4096, d_{\text{out}} = 1024$ for

```
(AB_emb): ParameterDict(
    (q_proj): Object of type: ParameterDict
    (v_proj): Object of type: ParameterDict
  (q_proj): ParameterDict(
      (A): Parameter containing: [torch.cuda.FloatTensor of size 128]
      (B): Parameter containing: [torch.cuda.FloatTensor of size 128]
  )
  (v_proj): ParameterDict(
      (A): Parameter containing: [torch.cuda.FloatTensor of size 128]
      (B): Parameter containing: [torch.cuda.FloatTensor of size 128]
  )
)

(rank_emb): Sequential(
    (0): Embedding(8, 128)
    (1): LayerNorm((128,), eps=1e-05, elementwise_affine=True)
)
```

Listing 3: Detailed A/B and rank embedding of `T2L`.

`v_proj` module. $r = 8$ for all adapters in this work. Finally, we list the number of trainable parameters of each architecture: $55,252,992$ for **L** , $34,282,240$ for **M** , $4,923,392$ for **S** , $3,407,872$ for LoRA.

## C   HYPERLORA INTIALIZATION

We use *Bias-HyperInit* (Beck et al., 2023) to initialize **L** **T2L**. Bias-HyperInit initializes the linear output head of the hypernetwork such that the weights are all zero and the bias matches the initialization of the underlying layers. In our work, this corresponds to the output bias of the **L** hypernetwork being initialized to $U(-\frac{1}{d}, \frac{1}{d})$ for the $A$ head and all zero for the $B$ head to match the initialization of traditional LoRA. For other architectures, we aim to match the gradient magnitude to **L** at the beginning of training. That is, for **M** architecture, we initialize the bias of the output head to $U(-\frac{1}{\sqrt{2}d}, \frac{1}{\sqrt{2}d})$. Finally, **S** output bias is initialized to $U(-\frac{1}{\sqrt{r2d}}, \frac{1}{\sqrt{r2d}})$. Without this explicit hypernetwork initialization, the training is unstable, and often leads to failed training runs.

## D   TRAINING DETAILS

All models trained in this work fit in a single H100 GPU (80GB of VRAM). Notably, SFT requires much more memory because of the need to backpropagate the gradient through the base LLM. Reconstruction training, on the other hand, should be possible in a modern consumer-grade GPU.

For reconstruction training, we fix the training epochs to be 100K but scale the batch size to match the number of target LoRA adapters. This means the model trains much faster for a lower number of target LoRAs while maintaining the same number of optimizer steps. For reference, training to reconstruct 9 benchmark-specific LoRAs takes around 10 minutes to complete, while training to reconstruct 489 SNI LoRA adapters takes around 10 hours.

For SFT training with fixed compute budget, we aim to keep the number of optimizer steps the same as we do for reconstruction training. However, since we cannot fit all fine-tuning samples, we scale the number of epochs inverse to the number of training tasks.

Additionally, for reconstruction training, instead of predicting the weights directly, `T2L` learns to predict the z-score of a normal distribution of each weight entry in the low-rank $A, B$ matrices. At test time, the output is multiplied by the standard deviation of each element before adding to the mean, converting the prediction to the correct scale.

## E   TRAINING AND EVALUATION DATASETS

We use 500 SNI datasets publicly available at https://huggingface.co/Lots-of-LoRAs. 489 tasks are used for training and the rest for evaluation. Specifically, we use the following evaluation

tasks: task_035, task_039, task_1557, task_202, task_304, task_362, task_614, task_701, task_706, task_710, task_726. For the in-context learning baseline, we use 3-shot in-context examples taken from the training split of each benchmark except MBPP that has an explicit split for in-context prompting. HumanEval only has the test split, therefore it is always evaluated against in the zero-shot manner.

### E.1 BENCHMARK DETAILS

Every benchmark used in the experiments is publicly available in HuggingFace dataset space. We evaluate the models on the benchmarks detailed as follows.

#### E.1.1 GSM8K

We evaluate the models on the test split, using chain-of-thought response pre-filling: *"Let's think step by step."*

#### E.1.2 HUMANEVAL AND MBPP

We use the `evalplus` library (Liu et al., 2023) for coding evaluation. For both MBPP and HumanEval, we use the following response pre-fill: *"""`python"*

### E.2 QUESTION-ANSWERING TASKS

The rest of the benchmarks are question-answering based tasks. In these tasks, we do not use response-prefilling. Instead, each task has a specific instruction template shown in Listing 4.

## F UTILIZING FULL ADAPTATION MATRIX VS LOW-RANK MATRICES

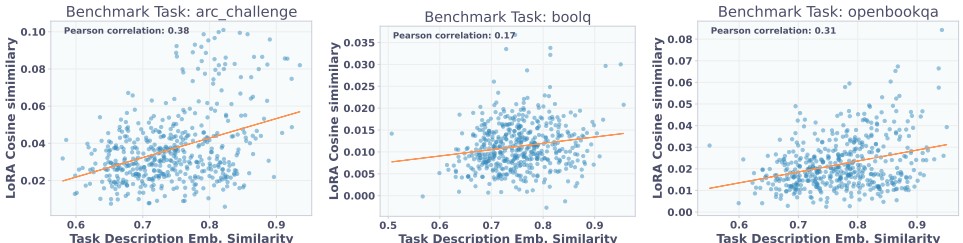

Figure 6: Each plot shows the similarity between a benchmark LoRA adapter and 489 SNI-trained adapters in the $\Delta W$ weight space. There is a positive correlation between the two variables indicated by small positive Pearson correlation coefficients.

Similar to Fig. 4, Fig. 6 show the similarity of SNI adapters to benchmark-specific adapters, but instead of using the concatenation of flattened $A$ and $B$ matrices, we use flattened $\Delta W$ instead. With the change, we find a positive correlation between the task embedding similarity and the adapter similarity in the weight space. This is likely because, for a given $\Delta W$ matrix, there are many possible permutations of low-rank matrices $A$ and $B$. This suggests that if we compute the reconstruction loss in the full adaptation matrix space, reconstruction-trained `T2L` could generalize better. However, we empirically find that it does not outperform `T2L` trained to reconstruct low-rank matrices at zero-shot LoRA generation.

~~Furthermore, we also ablate how the prediction offset is applied to `T2L` under SFT training. First, the offset could be added to the low-rank matrices:~~

$$\Delta W = (B_{\text{pred}} + B_{\text{offset}})^T (A_{\text{pred}} + A_{\text{offset}}) \qquad (6)$$

$$= B_{\text{pred}}^T A_{\text{pred}} + B_{\text{pred}}^T A_{\text{offset}} + B_{\text{offset}}^T A_{\text{pred}} + B_{\text{offset}}^T A_{\text{offset}}. \qquad (7)$$

~~The other approach is to apply the offset to the full adaptation matrix:~~

$$\Delta W = B_{\text{pred}}^T A_{\text{pred}} + B_{\text{offset}}^T A_{\text{offset}} \qquad (8)$$

```
OQA_TEMPLATE = (
    "Complete the following passage or answer the question by choosing the correct choice.\n\n"
    "{question_stem}\n\n"
    "{choices[label][0]}: {choices[text][0]}\n{choices[label][1]}: {choices[text][1]}\n"
    "{choices[label][2]}: {choices[text][2]}\n{choices[label][3]}: {choices[text][3]}\n\n"
    "You must respond with the letter corresponding to the correct choice (A,B,C,D) without any explanation."
)
ARC_TEMPLATE = (
    "Answer the question below by choosing the correct choice.\n\n"
    "{question}\n\n"
    "{choices[label][0]}: {choices[text][0]}\n{choices[label][1]}: {choices[text][1]}\n"
    "{choices[label][2]}: {choices[text][2]}\n{choices[label][3]}: {choices[text][3]}\n\n"
    "You must respond with the letter corresponding to the correct choice without any explanation."
)
HSWAG_TEMPLATE = (
    "You are provided with an incomplete passage below as well as 4 choices of continuation "
    "with only one of them being the correct ending. "
    "Treat the endings as being labelled 0, 1, 2, 3 in order.\n\n"
    "Passage: {ctx}\n\n"
    "0: {endings[0]}\n"
    "1: {endings[1]}\n"
    "2: {endings[2]}\n"
    "3: {endings[3]}\n\n"
    "You must respond with the only number corresponding to the correct ending (0,1,2,3) for the passage "
    "without any explanation."
)
PIQA_TEMPLATE = (
    "Choose the option that either answers the question, completes the sentence, or solves the problem. "
    "Pay attention to the properties of the objects in the question and how they interact with each other. "
    'If both options are correct, choose the one that is more convenient or more common.\n\n"""{goal}"""\n\n'
    "0: {sol1}\n1: {sol2}\n\n"
    "You must respond with either `0` or `1` without any explanation."
)
WINOGRANDE_TEMPLATE = (
    "Given the following situation:\n\n{sentence}\n\nWhich option is correct?\n\n"
    "Option 1: {option1}\n\nOption 2: {option2}\n\n"
    "You must respond with either `1` or `2` without any explanation."
)
BOOLQ_TEMPLATE = (
    "{passage}\n\nQuestion: {question}?\n\nPlease answer with either `true` or `false` without any explanation."
)
```

Listing 4: Instruction templates of QA-based benchmark tasks.

In a preliminary experiment, we find that applying the offset to the low-rank matrices performs better than the alternative for SFT training. We hypothesize that the cross terms (the middle two terms in Eq. (7)) ease the learning process. Effectively, the offset matrices in the cross terms act as 'answer bases' for $A_{\text{pred}}$ and $B_{\text{pred}}$ to act upon.

# G  GENERATING TASK DESCRIPTIONS WITH A FOUNDATION LANGUAGE MODEL

---

**System message**

You are a creative and helpful assistant.

---

**Prompt**

Given the following question-response pairs, please give a short description of the task describing what the task is.

{IN CONTEXT EXAMPLES}

Now, you must describe the task based on the following question-response pairs.

{5 sampled question-answer pairs}

Please use the information in the question-answer pairs and example description and come up with several descriptions that explain the task. Each description should be written in plain text, with the following format.

Description 1: DESCRIPTION_1
Description 2: DESCRIPTION_2
...

You should also be creative and vary the structure and the length of the descriptions such that they'll be diverse and cover various writing styles. You should ignore the specific question-answer pairs and focus on the high-level concept and topic of the task in general.
**DO NOT** describe that there are multiple choice options or the format of the answer.
**DO NOT** include the answer format, e.g., 'choose the correct option', 'answer with only one word', etc.
**DO NOT** describe how to answer the question, but rather what the task is about and the skills and knowledge required.
You can include reasoning steps that should be used to reach the expected answer.

Response with 20 descriptions. Use simple words and please be clear and diverse in your descriptions.

---

**In-context examples**

Here are some examples of the structure of the task of describing a task based on question-response pairs.

## Example question-answer pair: 1
### Input
You are given a question on high school macroeconomics. You are also given 4 answer options (associated with 'A', 'B', 'C', 'D'), out of which only one is correct. You need to answer the question by selecting the correct option. You should only answer with the choice letter, not the whole answer.
Input: Allocative efficiency (A)means that no inferior products will be produced. (B)implies that the economy's output is distributed evenly. (C)means that those who work hardest will get more. (D)implies that resources are used to produce the goods and services society desires in just the right amounts.
Output:
### Expected output
D
### Plausible descriptions
Description 1: Your job is to analyze the provided question about economics. Use your understanding of economic principles to guide your choice.
Description 2: Utilize your economic understanding to determine which choice is right. The correct answer will be the one that best aligns with economic principles.

## Example question-answer pair: 2
### Input
In this task, you are given a country name and you need to return the capital city of the given country.
Input: Senegal
Output:
### Expected output
Dakar
### Plausible descriptions
Description 1: Given the name of a country, your job is to provide its capital city.
Description 2: For each country listed, determine and state its capital city. This requires familiarity with global locations and capitals.

---

Figure 7: The prompt template used to query `GPT-4o mini` for task descriptions.

We automate task description generation for each task by leveraging powerful closed-source language models (Achiam et al., 2023). We query `GPT-4o mini` with carefully constructed prompts that incentivize diversity to facilitate downstream generalization. In particular, we generate 200 descriptions per task by querying the model 10 times, each time asking for 20 descriptions given randomly sampled five question-answer pairs from the task. We leverage in-context learning by providing examples of question-answer pairs with matching descriptions. Finally, we also designed our prompts to avoid overly verbose responses and unnecessary information, such as explicit mentions of answer formats and additional instructions. We use the generated descriptions for the training and benchmark tasks. Fig. 7 shows the exact prompt used for querying `GPT-4o mini` for task descriptions.

## H  EXAMPLE OF TASK DESCRIPTIONS

Here, we provide examples of task descriptions used in the experiments.

---

**Training descriptions**

**sni_cosmosqa_passage_inappropriate_binary**
- Assess whether the given passage contains any elements that are unsuitable or illogical. Contextual understanding is key to making your evaluation.
- Look closely at the information provided in the context and determine its appropriateness or nonsensical nature based on logical reasoning.
- Assess given contexts critically, marking whether they hold inappropriate content or convey meaning in a way that is difficult to comprehend.

**sni_winomt_classification_gender_identifiability_anti**
- In this task, you will distinguish between identifiable and unidentifiable gender references in sentences featuring different professions.
- Your task consists of evaluating professional descriptions within sentences and determining if their respective genders can be classified as clearly identifiable or obscure.
- Engage with sentences that present two different professions, paying attention to pronouns that could reveal or obscure the gender of the highlighted role.

**sni_kth_largest_element**
- In this task, you are required to dissect a set of integers and identify which one corresponds to the kth position when sorted by size. Knowledge of ascending order and magnitude awareness are pivotal.
- Your mission here is to discover which number holds the kth place when considering size among others in a list. Practicing sorting and prioritization will be beneficial.
- The job is to pick out the kth greatest number from a list of integers, which means reevaluating them according to their increasing or decreasing order.

---

Figure 8: Examples of training descriptions from three SNI training tasks.

**Evaluation descriptions**

**boolq**
- Analyze the given details about various subjects, including movies, sports, and television shows. Your role is to confirm whether certain claims are true or false.
- Your task is to determine the truthfulness of specific statements based on the provided background information. This requires careful reading and comprehension of the content.
- The goal is to evaluate factual claims made in relation to highlighted texts. You will need to discern whether the statements align with the information provided.

**gsm8k**
- You will be tasked with interpreting mathematical situations described in words. The goal is to use logical reasoning and calculations to determine the numerical answers based on the context provided.
- This task challenges your problem-solving abilities through mathematical reasoning. You must carefully read each scenario and systematically work through the data to compute the final outcome.
- Your role is to engage with practical math scenarios presented as questions. The task requires translating textual data into numerical operations that will lead you to the final solution.

**humaneval**
- Engage in building distinct functions that meet the requirements of various presented problems, honing your ability to translate problem statements into logical code. Utilize structured thinking to implement efficient solutions.
- You are tasked with generating specific solutions in Python by interpreting problem descriptions associated with tasks like counting odds or validating inputs. Recognizing patterns and leveraging programming techniques will be beneficial.
- This task focuses on developing algorithms in Python for specific scenarios, such as counting characters, assessing conditions between numbers, or converting integers into a different format. Critical thinking and algorithmic design will be important.

Figure 9: Task descriptions of the benchmark tasks: boolq, gsm8k, and humaneval.

**Evaluation descriptions**

**mbpp**
- Your challenge is to solve a series of problems by writing functions in Python. These problems require handling lists and strings, allowing you to showcase your proficiency in coding while addressing practical programming scenarios.
- You will be tasked with creating various Python functions that tackle programming challenges. The exercises will test your ability to manipulate data structures, search for patterns, and implement checks on numerical products.
- The goal is to develop Python functions that perform designated operations on lists and strings. This requires a solid grasp of logical reasoning and the ability to apply relevant algorithms in your code.

**winogrande**
- In this exercise, you need to read short narratives and discern which person or object fits best within the context of the sentence.
- This task requires synthesizing information from concise textual scenarios to identify crucial elements that drive the narrative forward.
- The goal is to evaluate descriptions and select the entity that best aligns with the sentiments or actions presented in the scenario.

**piqa**
- You will explore practical questions and select an answer that presents a logical and widely accepted approach to solve a given problem or complete a task successfully.
- Analyze the provided scenarios where practical advice or solutions are required, focusing on selecting the most commonly used or convenient method.
- Given a question related to common tasks, your responsibility is to discern which proposed solution aligns with typical practices or makes the task easier to achieve.

Figure 10: Task descriptions of the benchmark tasks: mbpp, winogrande, piqa

**Evaluation descriptions**

**hellaswag**
- This task revolves around completing an unfinished text by selecting an ending that matches its tone and context. It requires you to think critically about how narratives develop and conclude effectively.
- This task asks you to select a suitable conclusion for an unfinished narrative or instructional content. It tests your comprehension and reasoning skills as you assess how well each option aligns with the given text.
- Your task involves completing an incomplete passage by selecting the ending that logically continues the context provided. This requires reading comprehension and the ability to infer meaning from a text.

**arc_easy**
- Your job is to discern which information best answers a posed question, focusing on practical examples and scientific principles. This requires a strong grasp of underlying concepts in ecology or physics.
- You will analyze questions that explore important connections such as environmental issues or animal adaptations. Utilize your background knowledge to evaluate and select the most fitting answer.
- This task involves selecting answers that reflect accurate relationships or effects seen in nature or society. You will need to sort through potential choices critically to find the appropriate one.

**arc_challenge**
- This task is about analyzing questions which examine your grasp of scientific ideas. You must connect conceptual knowledge with practical examples from geology, ecology and environmental changes.
- The objective here is to evaluate various scientific scenarios and infer the most logical explanations or definitions based on established knowledge. This task will strengthen your analytical and reasoning skills in the context of natural science.
- Your role is to interpret questions focusing on earth science and biological interactions. This demands a clear understanding of relevant processes, such as decomposition, weathering, and species adaptation.

Figure 11: Task descriptions of the benchmark tasks: hellaswag, arc_easy, arc_challenge

**Evaluation descriptions**

**openbookqa**
- Analyze the provided statements carefully and determine which one best fits into the context of the passage. This requires comprehension skills and the ability to make logical inferences.
- Consider each option in relation to what is presented in the input. Discern which one logically completes or responds accurately to the notion being expressed.
- Here, you'll be presented with different statements, and your role is to decide which one appropriately complements or responds to a scenario. This process involves critical analysis and synthesis of information.

Figure 12: Task descriptions of the benchmark tasks: openbookqa

**Random descriptions**

- dogs;cats;bananas;
- 7@9.qwepra#/.sd,s'2OC^O39u#rdagjbL
- ggggggggggggggggggggggg

Figure 13: Random descriptions

## I    SCALING NUMBER OF DESCRIPTIONS PER TASK

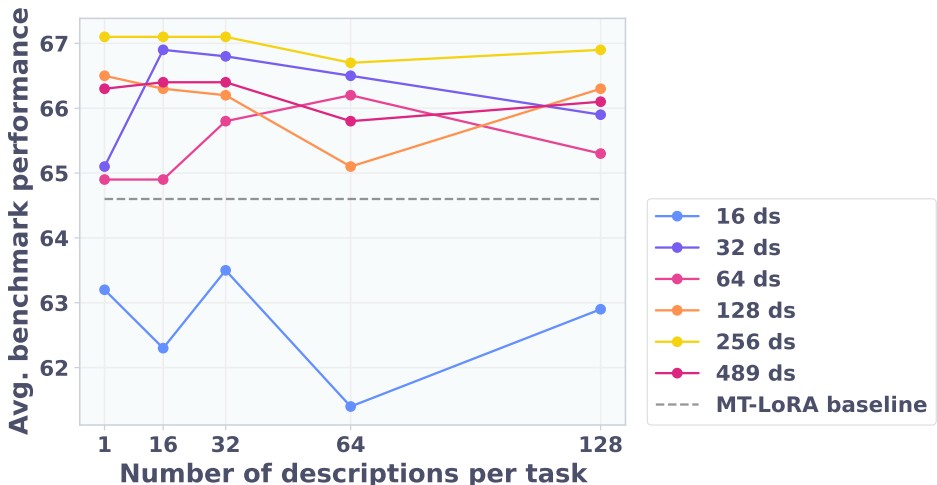

Figure 14: Zero-shot benchmark performance of SFT-trained **T2L** with varying number of descriptions per training task.

Fig. 14 shows mixed results on the benchmark performance when vary the number of descriptions per training task. For consistency, we always train **T2L** with 128 descriptions per training task.