# OpenReview forum: "Instant Transformer Adaption via HyperLoRA"
_ICLR.cc/2025/Conference — Submitted to ICLR 2025_

### Official Review · Reviewer_jXry · 2024-10-30

**Soundness:** 3
**Presentation:** 2
**Contribution:** 2
**Rating:** 5
**Confidence:** 4

**Summary:**

This work proposes using hypernetworks to generate LoRA weights for unseen tasks based on natural language instructions. It does so by embedding the instructions using an existing embedding model (or activations from an LLM) and generates the LoRA weights using a newly trained neural network. The authors show that this approach to generate LoRA weights can extend to tasks not seen in the training set.

**Strengths:**

- The authors provide clear evaluation of their proposed method.
- The authors conduct informative analysis on different configurations for training the hypernetwork, showing the impact of varying the number of tasks and embedding method.

**Weaknesses:**

- The authors omit key prior work [e.g. 1,2,3] in their related work. For instance, [1] similarly uses SNI and trains a hypernetwork to generate LoRAs, with both few-shot as well as instruction-only configurations. [3] Likewise also uses both SNI (formerly NIv2) and an instruction-only setting.
- Correspondingly, the authors largely fail to compare their currently method to existing hypernetwork methods (both the ones omitted above, as well as the ones they cite in their work). The primary other method they compare to is Arrow Routing, which seems largely unrelated to the setting besides the connection of (directly rather than indirectly) relying on a set of pretrained LoRAs.
- The results are directionally good but many of the evaluated tasks already look saturated. There are also some oddities such as LoRA on PiQA underperforming the base model, which seems to suggest an inadequate configuration for their baseline.
- Parts of the writing are unclear, e.g. the distinction between the "Task Description Embeddings" setting and the unseen task descriptions used for "Zero-Shot LoRA Generation". (My understanding is that all task descriptions on evaluated tasks should be unseen?) There is also reference to both Train and Eval descriptions referenced in Table 5, which makes it further unclear what tasks/descriptions are actually unseen during training.
- There is some unconvincing post-hoc justification for empirical results. For instance "This result is in line with the general knowledge that certain inductive biases improve models' robustness and generalization" is used to explain why the middle M configuration outperform S and L reads to me like trying to force a scientific interpretation to an unexplained (which is okay) quirk in empirical result.

[1] HyperTuning: Toward Adapting Large Language Models without Back-propagation
[2] Learning to Compress Prompts with Gist Tokens
[3] Boosting Natural Language Generation from Instructions with Meta-Learning

**Questions:**

- See above: what is the distinction between seen and unseen task descriptions?
- How are the one-hot embeddings generated?

---

> ### Author Response · Authors · 2024-11-22
>
> We thank the reviewer for their time, feedback and questions. We appreciate the depth of the raised points and are glad to hear that our work contains clear evaluation of the proposed method and informative analysis on different configurations for training the hypernetwork.
>
> Here, we address the concerns and questions raised by the reviewer.
>
> > The authors omit key prior work [e.g. 1,2,3] in their related work. For instance, [1] similarly uses SNI and trains a hypernetwork to generate LoRAs, with both few-shot as well as instruction-only configurations. [3] Likewise also uses both SNI (formerly NIv2) and an instruction-only setting.
>
> We thank the reviewer for pointing out missing key prior work. We are very much dedicated to providing the right degree of background and giving credit where credit is due. Adding to the reviewer list of missing prior work, we additionally found HINT [4]. We have updated the background section and provided some additional clarification here:
>
> - We use frontier instruction fine-tuned model as the base models, i.e., Mistral-7B-Instruct-v0.2, Gemma-2-2b-it and Llama-3.1-8B-Instruct (added in **Section 4.3**), as opposed to prior work [1,3,4] that heavily focuses on pre-trained encoder-decoder, e.g., T5 [5] or BART [6].
> - [1,3,4] rely on initializing part of their hypernetwork from the base model to achieve good performance or stable training as opposed to ours, which is task-embedder agnostic and can freely change the task embedding model (**Section 5.4**).
> - We also explore the use of reconstruction loss with the purpose of distilling knowledge of task-specific LoRAs into the hypernetwork and compare that to SFT-trained hypernetwork. We empirically find that reconstruction-trained HyperLoRA cannot generalize well compared to SFT-trained ones. Furthermore, we provide compelling evidence as to why that is the case. Thus, this work provides crucial insight into the training regime of the hypernetwork for fast adaptation.
> - Additionally, our work utilizes generated descriptions from GPT-4o instead of the ones provided by the SNI dataset. We have added an ablation study on the impact of the training description source (**Section 5.3**)
>
> We have updated the related work section accordingly. We hope that this addresses the reviewer’s concern about missing prior work.
>
>
> > Correspondingly, the authors largely fail to compare their current method to existing hypernetwork methods (both the ones omitted above, as well as the ones they cite in their work). The primary other method they compare to is Arrow Routing, which seems largely unrelated to the setting besides the connection of (directly rather than indirectly) relying on a set of pretrained LoRAs.
>
> We agree with the reviewer that we missed important hypernetwork-based baselines. As our work shares many ideas with prior work, the main distinction of our work is the use of frontier base models and the challenging evaluation setup.
> To address the reviewer’s concern regarding hypernetwork-based baselines, we added Hyperdecoders [7] (**Table 2**). Our HyperLoRA method compares favorably.
>
> Furthermore, we also added few-shot in-context learning and task description prepending as baselines (**Table 2,3,4**).
> Another distinction of our work is the use of generated descriptions. Thus, we have added another ablation study (**Section 5.3**) to compare HyperLoRA trained on generated task descriptions vs task definitions provided from SNI datasets. This baseline should be a representation of many prior works that rely on the task description from the SNI dataset.
> We hope these additions address the reviewer’s concern regarding the baselines.
>
> > The results are directionally good but many of the evaluated tasks already look saturated. There are also some oddities such as LoRA on PiQA underperforming the base model, which seems to suggest an inadequate configuration for their baseline.
>
> There is a significant gap between multi-task LoRA and task-specific LoRA baseline across the benchmark tasks, indicating that there is room for improvement for the benchmarks considered in this work. HyperLoRA consistently outperforms the multi-task baseline, closing the gap between the multi-task LoRAs and task-specific LoRAs. Regarding the PIQA task, we did try retraining the PiQA LoRA with longer training and early stopping but still got a similar result. The authors of the PIQA dataset also mentioned in their paper that most finetuning configurations tend to be unstable because of the nature of the dataset.
>
> We have also included two more base models (**Section 4.3**) and HyperLoRA consistently outperforms the baselines across various model families and sizes. We hope this additional experiment addresses the reviewer’s concern regarding saturated performance and the robustness of our proposed method.

---

> > ### Author Response · Authors · 2024-11-22
> >
> > > Parts of the writing are unclear, e.g. the distinction between the "Task Description Embeddings" setting and the unseen task descriptions used for "Zero-Shot LoRA Generation". (My understanding is that all task descriptions on evaluated tasks should be unseen?) There is also a reference to both Train and Eval descriptions referenced in Table 5, which makes it further unclear what tasks/descriptions are actually unseen during training.
> >
> > We thank the reviewer for pointing out unclear sections of the paper.
> >
> > In **Table 1**, all hypernetworks are trained on the benchmark tasks to test their compression capability. Thus, the benchmarks are seen during training. We explore two embedding types in this setting: one-hot task embeddings and task description embeddings.
> >
> > In **Table 5**, we probe reconstruction-trained hypernetworks trained and tested on 9 benchmark tasks. Therefore, in this experiment, the benchmark tasks are seen during training and we vary the description types after training to construct Table 5. Train descriptions are the descriptions that are used during training and are aligned with the tasks. Eval descriptions are also aligned with the tasks but are unseen.
> >
> > We added more explanation to the text in **Section 4.1** as follows:
> > *“We note that the benchmark tasks are indirectly seen during training by HyperLoRA as it learns to distill benchmark-specific LoRAs.”*
> >
> > We also added more explanation to the text in **Section 5.5** as follows:
> > *“Note that we use reconstruction-trained HyperLoRA in this experiment. That is, the hypernetwork has seen training descriptions of the benchmarks during training.”*
> >
> > > There is some unconvincing post-hoc justification for empirical results. For instance "This result is in line with the general knowledge that certain inductive biases improve models' robustness and generalization" is used to explain why the middle M configuration outperform S and L reads to me like trying to force a scientific interpretation to an unexplained (which is okay) quirk in empirical result.
> >
> > We thank the reviewer for the comment. We have removed this part.
> >
> > > See above: what is the distinction between seen and unseen task descriptions?
> >
> > In **Table 5**, we probe the reconstruction-trained hypernetwork trained and tested on 9 benchmark tasks. Therefore, in this experiment, the tasks are all seen and we vary the description types. Train descriptions are the descriptions that are used during training and are aligned with the tasks. Evaluation descriptions are also aligned with the tasks but are unseen.
> >
> > > How are the one-hot embeddings generated?
> >
> > One-hot embeddings are 1D vectors with the size of the number of training tasks. They are only used for reconstruction training to test the compression capabilities of hypernet. Hypernetwork trained this way cannot generalize beyond training tasks. Only Hypernetworks in Table 1 with one-hot embeddings are trained this way.
> >
> > We thank the reviewer again for spending their time to provide constructive feedback. We hope to have addressed the reviewer’s remaining concerns. We would highly appreciate it if the reviewer could confirm that the concerns have been addressed. If so, we’d like to ask the reviewer to consider increasing the recommendation score. We are happy to discuss further if any additional questions arise. We are looking forward to your response.
> >
> > Best wishes,
> >
> > The authors
> >
> > - [1] HyperTuning: Toward Adapting Large Language Models without Back-propagation https://arxiv.org/abs/2211.12485
> > - [2] Learning to Compress Prompts with Gist Tokens https://arxiv.org/abs/2304.08467
> > - [3] Boosting Natural Language Generation from Instructions with Meta-Learning https://arxiv.org/abs/2210.11617
> > - [4] HINT https://arxiv.org/pdf/2212.10315
> > - [5] Exploring the Limits of Transfer Learning with a Unified Text-to-Text Transformer. https://arxiv.org/pdf/1910.10683
> > - [6] BART: Denoising Sequence-to-Sequence Pre-training for Natural Language Generation, Translation, and Comprehension https://arxiv.org/abs/1910.13461
> > - [7] Hyperdecoders: Instance-specific decoders for multi-task NLP https://arxiv.org/pdf/2203.08304

---

> > > ### Author Response · Authors · 2024-11-26
> > >
> > > Dear Reviewer jXry,
> > >
> > > We wanted to kindly remind you that the discussion period for the review process is coming to an end soon. We kindly ask if the reviewer could consider whether our clarifications and additional experiments support an increase in their score. We are happy to discuss further if any additional questions arise. We are looking forward to your response.
> > >
> > > Thank you once again for your time and feedback.
> > >
> > > Best regards,
> > >
> > > Authors

---

> > > > ### Author Response · Authors · 2024-12-03
> > > >
> > > > Dear Reviewer jXry,
> > > >
> > > > We would like to gently remind you that the rebuttal period for the review process will conclude soon. We truly appreciate the time and effort you've already dedicated to reviewing our submission. Consequently, we have significantly improved and revised the manuscript including new experiments, baselines, and clarifications (see our general response to all reviewers).
> > > >
> > > > We would appreciate it if the reviewer could confirm whether the concerns have been addressed. Additionally, if the reviewer's concerns have been addressed, we would like to ask the reviewer to revise the rating score accordingly.
> > > >
> > > > If the reviewer has any further questions or concerns regarding our clarifications or additional experiments, please don't hesitate to reach out. We are eager to address any remaining points that might influence your final assessment.
> > > > Thank you once again for your valuable feedback and your continued support.
> > > >
> > > > Best regards,
> > > >
> > > > Authors

---

### Official Review · Reviewer_M5pM · 2024-11-03

**Soundness:** 3
**Presentation:** 3
**Contribution:** 3
**Rating:** 5
**Confidence:** 3

**Summary:**

The paper focuses on the problem of reusing knowledge from pretrained parameter-efficient adapters to new tasks without target task specific fine-tuning.

In particular, the authors propose to leverage the hypernetwork as a means to generate such LoRA adapters by using two kinds of training signals, task-specific training loss and task-dependent LoRA weight reconstruction loss.
Specifically, the hypernetwork generates three variants of LoRA matrices by using information of the target task (one-hot or natural language embeddings) and target module to be adapted (depth and FFN vs MHA). Compared with the backbone LM, the hypernetwork is parameter-efficient too.

The authors then apply the proposed method on top of a pretrained language model (Mistral-7B-instruct) with several representative English natural language understanding tasks.
Compared with LoRA baselines and recent work on combining LoRA weights for unseen tasks, the proposed method HyperLoRA shows promising improvements.

**Strengths:**

The paper studies a practically interesting problem, i.e., adapting LLMs on the fly based on the natural language descriptions.

The proposed HyperLora architectures are well designed.

Experiments show positive results of the proposed approach.

**Weaknesses:**

The experiments only use one base model (i.e., Mistral-7B-Instruct). It is unclear whether the approach can generalize to other model families (e.g., Llama, Phi) and other model sizes.
It would be also interesting to see if the HyperLora can benefit from learning to generate LoRA adapters for different model families.

It would be useful to add few-shot and many-shot in-context-learning results as baselines as well. And also compare the cost between such in-context-learning and the proposed HyperLoRA.

The need of using a multi-task LoRA as a prediction offset to boost the performance is a little undesired, as it requires extra cost for training a multi-task LoRA in the first place.

Minor:
Line 225: it would be useful to describe the prediction offset clearly in the main body of the paper, or at least refer to the Eq 6 & 7 in the appendix.

**Questions:**

1) How stable is training process of the proposed method? e.g., different AB configs, different batch sizes are required for different configs.

2) Is there any benefits of combining the SFT loss and the reconstruction loss?

3) What is included in the task descripton? Just the high-level task description? What about adding a few examples of the tasks?

---

> ### Author Response · Authors · 2024-11-22
>
> We thank the reviewer for their thoughtful comments and appreciation. We also thank the reviewer for commenting on the hypernetwork architectures as being **well designed** and that our **experiments show positive results of the proposed approach**. We are also glad to hear that the paper studies a **practically interesting problem**.
>
> We would like to take the opportunity to discuss their suggestions and questions.
>
> > The experiments only use one base model (i.e., Mistral-7B-Instruct). It is unclear whether the approach can generalize to other model families (e.g., Llama, Phi) and other model sizes. It would be also interesting to see if the HyperLora can benefit from learning to generate LoRA adapters for different model families.
>
> We have included more base models, Gemma-2-2b-it and Llama-3.1-8B-Instruct, in the experiments (**Section 4.3**). We observe that HyperLoRA consistently outperforms the baselines across all tested models with varying model sizes and architectures. We note that all HyperLoRA instances are trained with the same set of hyperparameters across base models. This result suggests the robustness of HyperLoRA to different model families and sizes.
> We also thank the reviewer for suggesting the idea of transferring between base models. It is conceivable that doing so could have a positive impact on the performance and generalization of HyperLoRA. However, different model families might have different architectural choices (e.g., grouped attention). Not to mention that their latent spaces and weights could have arbitrary permutations, making interoperability of HyperLoRA much more difficult. Tackling these challenges would require a significant amount of engineering and potentially new techniques altogether. We leave this investigation for future work.
>
> > It would be useful to add few-shot and many-shot in-context-learning results as baselines as well. And also compare the cost between such in-context learning and the proposed HyperLoRA.
>
> We thank the reviewer for the suggestion. We have added these baselines to the experiment (**Table 2,3,4**). HyperLoRA consistently outperforms the in-context learning baseline across all base models.
>
> The evaluation cost (i.e., memory and speed) of in-context learning is known to be much higher than using only the query itself. Additionally, the extra cost is applied to all queries in each benchmark, which can be prohibitive for large datasets. In contrast, evaluating HyperLoRA requires only one single-forward pass, which has negligible cost compared to the number of queries, to generate a LoRA adapter. LoRA generation cost is then amortized over the number of queries as opposed to proportional in the case of in-context learning.
>
> > The need to use a multi-task LoRA as a prediction offset to boost the performance is a little undesired, as it requires extra cost for training a multi-task LoRA in the first place.
>
> Thank you to the reviewer for this valid observation. Previously, we used multi-task LoRA as the prediction offset to boost performance. However, we discovered that training for longer without the offset increases performance over hypernetworks that use prediction offset. Hence, we remove the mention of the prediction offset from the paper.
>
> > Minor: Line 225: it would be useful to describe the prediction offset clearly in the main body of the paper, or at least refer to Eq. 6 & 7 in the appendix.
>
> Since we discovered that training for longer without the offset increases performance over hypernetworks that use prediction offset, we have removed the offset prediction and updated the text accordingly. We are sorry for the confusion caused by this.

---

> > ### Author Response · Authors · 2024-11-22
> >
> > > How stable is training process of the proposed method?
> >
> > The training is stable thanks to proper hypernetwork weight initialization. We have added the weight initialization detail in **Appendix C** as follows:
> >
> > *“We use Bias-HyperInit to initialize L HyperLoRA. Bias-HyperInit initializes the linear output head of the hypernetwork such that the weights are all zero and the bias matches the initialization of the underlying layers. In our work, this corresponds to the output bias of the L hypernetwork being initialized to $U(-\frac{1}{d}, \frac{1}{d})$ for the $A$ head and all zero for the $B$ head to match the initialization of traditional \lora{}.
> > For other architectures, we aim to match the gradient magnitude to L at the beginning of training. That is, for M architecture, we initialize the bias of the output head to $U(-\frac{1}{\sqrt{2}d}, \frac{1}{\sqrt{2}d})$. Finally, S output bias is initialized to $U(-\frac{1}{\sqrt{r2}d}, \frac{1}{\sqrt{r2}d})$. Without this explicit hypernetwork initialization, the training is unstable and often leads to failed training runs.”*
> >
> > >  e.g., different AB configs, different batch sizes are required for different configs.
> >
> > Regarding training of different output configurations, we use the same number of samples per batch across HyperLoRA architectures to ensure that all configurations have similar task diversity within a single batch. Additionally, we train HyperLoRA such that it produces full adaptation matrices for all layers and modules of the base model for each training sample.
> > For the L variation with batch size $bs$, number of layers $L$, number of modules $M$, task embedding size $d_t$, layer embedding size of $d_l$, module embedding size $d_m$, we have task embeddings with the shape [$bs, d_t$], which is batched over by the layer and module embeddings. This results in the input to the MLP of the hypernetwork with shape [$bs, M, L, d_m + d_l + d_t$], where each task embedding is repeated $M \times L$ times and concatenated to both the layer and module embeddings. Finally, the output of the hypernetwork has the shape of [$bs, M, L, 2, r, d$], where $M$ and $L$ are the learnable embeddings that are being batched on top of the samples in a single forward pass.
> >
> >
> > > Is there any benefits of combining the SFT loss and the reconstruction loss?
> >
> > We thank the reviewer for this suggestion. In fact, it has been done in concurrent work [1] to stabilize the training of the hypernetwork. In our work, we are able to train the hypernetwork stably without the reconstruction auxiliary loss, avoiding the necessity for training task-specific LoRAs in the case of SFT-trained HyperLoRA. We have added the hypernet initialization scheme that leads to stable training in **Appendix C**.
> >
> > Furthermore, we empirically found that reconstruction-trained HyperLoRA cannot generalize well to unseen tasks. Thus, we believe that using reconstruction auxiliary loss for SFT training would yield worse generalization, as also observed in [1] when the coefficient for the auxiliary loss is too high.
> >
> >
> > > What is included in the task descripton? Just the high-level task description? What about adding a few examples of the tasks?
> >
> > In our work, the generated task descriptions are intentionally high-level. This design prioritizes real-world scenarios in which users can interact with HyperLoRA by writing a task description within a few seconds. Including few-shot examples could help as natural language can be limited in specific tasks. However, for real-world scenarios, few-shot examples might not be as convenient for users to generate themselves compared to high-level task descriptions. We leave the investigation between ease-of-use and performance gain of few-shot examples to future work. Please see **Appendix G** for a set of generated task descriptions.
> >
> >
> > We thank the reviewer again for spending their time to provide constructive feedback. We hope to have addressed the reviewer’s remaining concerns. We would highly appreciate it if the reviewer could confirm that the concerns have been addressed. If so, we’d like to ask the reviewer to consider increasing the recommendation score. We are happy to discuss further if any additional questions arise. We are looking forward to your response.
> >
> > Best wishes,
> >
> > The authors
> >
> > [1] Hypernetworks in Meta-Reinforcement Learning. https://openreview.net/forum?id=N-HtsQkRotI

---

> > > ### Comment · Reviewer_M5pM · 2024-11-23
> > >
> > > Thanks the authors for providing more details.
> > >
> > > There is no further question from my side.

---

> > > > ### Author Response · Authors · 2024-11-23
> > > > **Revision of Score**
> > > >
> > > > Dear Reviewer M5pM,
> > > >
> > > > Thank you very much for your swift response. We are happy that we could address your points. We observed that your initial score seems to have decreased from 6 to 5. Was this by mistake? Is there anything we can do to improve the quality of our work and increase your score?
> > > >
> > > > We are happy to discuss further if any additional questions arise. We are looking forward to your response.
> > > >
> > > > Best wishes,
> > > > The authors

---

> > > > > ### Author Response · Authors · 2024-12-03
> > > > >
> > > > > Dear Reviewer M5pM,
> > > > >
> > > > > We would like to gently remind you that the rebuttal period for the review process will conclude soon. We truly appreciate the time and effort you've already dedicated to reviewing our submission.
> > > > >
> > > > > However, we are still unsure whether the rating change (from 6 to 5) was by mistake? Is there anything we can do to improve the quality of our work and increase your score?
> > > > >
> > > > > We are happy to discuss further if any additional questions arise. We are looking forward to your response.
> > > > >
> > > > > Thank you once again for your valuable feedback and your continued support.
> > > > >
> > > > > Best regards,
> > > > >
> > > > > Authors

---

### Official Review · Reviewer_SoVX · 2024-11-04

**Soundness:** 2
**Presentation:** 1
**Contribution:** 2
**Rating:** 5
**Confidence:** 3

**Summary:**

The paper proposes HyperLoRA, which is a method to generate a new low-rank adaptor(LoRA) based on the natural language task description without training an adaptor. Through experiments, the paper shows HyperLoRA successfully generate an adaptor for new task by achieving comparable or even higher performance than trained adaptor.

**Strengths:**

- Generating a new adaptor without tuning one can be utilized widely as it can minimize expensive tuning step.
- It is impressive that generating parameters itself rather than text or other data can show promising results

**Weaknesses:**

- The presentation of the paper lacks intuitiveness.
    - There are many typos and grammatical errors.
    - The figures are not clear.
        - For example, figure 1. left does not denote what the each arrow represents, and the figure looks like the HyperLoRA is optimizing both reconstruction loss and SFT loss at the same time, but it seems like the HyperLoRA optimize either one of the losses according to the paper
        - Also I was not able to find a part that shows the number of tasks in figure 3
    - Preliminary explanations for the many parts are not enough
        - For example, it does not explain what the prediction offset is.

**Questions:**

- What is the prediction offset?
- Is it possible to optimize both reconstruction loss and SFT loss at the same time?
- Does HyperLoRA work better than given task description as a prompt?

---

> ### Author Response · Authors · 2024-11-22
>
> We thank the reviewer for their time, feedback, and questions. We also thank the reviewer for commenting on the paper as **impressive at generating parameters with promising results**. We are also glad to hear from the reviewer that our work **can be utilized widely as it can minimize expensive tuning steps**.
>
> Here, we address the concerns and questions raised by the reviewer.
>
> > There are many typos and grammatical errors.
>
> We thank the reviewer for pointing this out. We have edited the text again and have corrected grammatical errors with the help of a writing assistant tool. If there are any remaining errors, we would love to fix them if the reviewer could provide us with specific details.
>
> > The figures are not clear. For example, figure 1. left does not denote what the each arrow represents, and the figure looks like the HyperLoRA is optimizing both reconstruction loss and SFT loss at the same time, but it seems like the HyperLoRA optimize either one of the losses according to the paper
>
> We have added a concise explanation of the arrows in the caption of Figure 1. They refer to the forward (blue) and backward (yellow) propagation of activities/gradients in the different settings. In the caption, we further explained that the HyperLoRA method is trained in one of the two settings (SFT or reconstruction). We hope this clarifies any confusion.
>
> > Also I was not able to find a part that shows the number of tasks in figure 3
>
> We have added additional information regarding the number of tasks in the caption of Figure 3.
>
> > Preliminary explanations for the many parts are not enough. For example, it does not explain what the prediction offset is.
>
> We are sorry that the explanation of the prediction offset was not included in the main text. The main explanation was included in **Appendix E** of the initial submission. However, we discovered that training for longer without the offset increases performance over hypernetworks that use prediction offset. Hence, we remove the mention of the prediction offset from the paper. We hope this removes any confusion around the prediction offset.
>
> > What is the prediction offset?
>
> We discuss the prediction offset in detail in **Appendix E** and **Section 5.4** of the initial submission. It refers to fixed matrices that are added to the generated LoRAs, which can be written mathematically as
>
> $$ \Delta {W} = (B_\text{pred} + B_\text{offset})^{T}(A_\text{pred} + A_\text{offset})  =  B_\text{pred}^{T}A_\text{pred} + B_\text{pred}^{T}A_\text{offset} + B_\text{offset}^{T}A_\text{pred} + B_\text{offset}^{T}A_\text{offset} $$
>
> Previously, we used a pre-trained multi-task LoRA as the prediction offset to boost performance. However, we discovered that training for longer without the offset increases performance over hypernetworks that use prediction offset. Hence, we removed the prediction offset from the paper and updated the text accordingly. We are sorry for the confusion caused by this.
>
> > Is it possible to optimize both reconstruction loss and SFT loss at the same time?
>
> We thank the reviewer for this suggestion. It has been done in concurrent work [1] to stabilize the training of the hypernetwork. In our work, we could train the hypernetwork stably without the reconstruction auxiliary loss, avoiding the necessity for training task-specific LoRAs in the case of SFT-trained HyperLoRA. We have added the hypernet initialization scheme that leads to stable training in **Appendix C**.
>
> Furthermore, we find empirically that reconstruction-trained HyperLoRA does not generalize well to unseen tasks. Thus, we believe that using reconstruction auxiliary loss for SFT training would yield worse generalization as also observed in [1] when the coefficient for the auxiliary loss is too high.
>
> > Does HyperLoRA work better than given task description as a prompt?
>
> We thank the reviewer for the suggestion. We have added this as a baseline and found that simple prompting improves the base model’s performance, but HyperLoRA still consistently outperforms this baseline (see **Table 2,3,4**).
>
>
> We thank the reviewer again for spending their time to provide constructive feedback. We hope to have addressed the reviewer’s remaining concerns. We would highly appreciate it if the reviewer could confirm that the concerns have been addressed. If so, we’d like to ask the reviewer to consider increasing the recommendation score. We are happy to discuss further if any additional questions arise. We are looking forward to your response.
>
> Best wishes,
>
> The authors
>
> [1] HyperloRA: Efficient cross-task generalization via constrained low-rank adapters generation. https://openreview.net/forum?id=xa4GYUSvhW

---

> > ### Author Response · Authors · 2024-11-26
> >
> > Dear Reviewer SoVX,
> >
> > We wanted to kindly remind you that the discussion period for the review process is coming to an end soon. We kindly ask if the reviewer could consider whether our clarifications and additional experiments support an increase in their score.
> > We are happy to discuss further if any additional questions arise. We are looking forward to your response.
> >
> > Thank you once again for your time and feedback.
> >
> > Best regards,
> >
> > Authors

---

> > > ### Comment · Reviewer_SoVX · 2024-11-28
> > > **Thank you for your response!**
> > >
> > > Most of my concerns are addressed. I will raise my score from 3 to 5. However, I believe the distinction between the related works claimed by the paper is a little insufficient to raise the score further.

---

### Official Review · Reviewer_tn5E · 2024-11-04

**Soundness:** 3
**Presentation:** 4
**Contribution:** 3
**Rating:** 6
**Confidence:** 3

**Summary:**

## Overview
The paper introduces HyperLoRA, which is a method involving training LLMs to adapt to new domains and tasks on-the-fly, without having to do expensive training at test-time. This is achieved through the use of a hypernetwork of LoRA adapters. During test-time, the natural language instruction is encoded and a LoRA adapter is zero-shot generated after a single forward pass.

The main hypothesis is that different LoRA adapters share similar underlying mechanisms and hence can be optimized simultaneously.

## Method + Experiments
- Three different variants (small, medium, large) with various output heads and learnable embeddings
- Can be trained either using LoRA reconstruction (shown to be poor at generalization) or through SFT
- 500 tasks. 11 held out for evaluation.
- Evaluate on 10 widely used tasks like ARC, HellaSwag, GSM8k, etc.
- Outperforms LoRA routing baseline

**Strengths:**

**1. Flexibility and efficiency with minimal overhead** -- The main strength of the method is that it allows for good generalization to new tasks without adding much extra in terms of training. It is also very flexible because it can adapt to unseen prompts during test time. This can help make LLMs a lot more accessible and easier to interact with.

**2. Strong results with a relatively simple method** -- The method generalizes well, even to zero-shot settings as seen in Table 2. Meanwhile, in Table 1, we see that In most cases, the model performs almost as well as, if not better, than the task-specific LoRA fine-tuning.

**3. Thorough ablations and analysis** -- The whole section 5 (ablations) and section 6 (analysis) go pretty in-depth into the models and what makes them work. For instance, the paper explores varying task embedding models, task descriptions, etc.

**4. Clean presentation** -- The paper uses color and space very effectively, which makes reading quite pleasant.

**Weaknesses:**

**1. Possible scaling concerns** -- I find it a bit concerning that adding more training tasks doesn't improve the performance (Table 3). Usually for most algorithms, adding more data would result in a better model. Otherwise, the concern is that the performance of the method will be capped at a certain level and it will be hard to increase further. Also, in the paper, the model was evaluated on 11 different tasks (which are somewhat close to each other). I am wondering how the model would scale to even more tasks beyond the basic ones.

**2. Comparison with full fine-tuning** -- The study is limited to the LoRA setting, which has been shown to perform worse than full fine-tuning, and this method likely wouldn't generalize to full fine-tuning settings. Similar to the above point, this makes me slightly concerned that the method might have its performance capped at a certain level.

**Questions:**

- The paper claims to introduce HyperLoRA but one of the papers cited in the paper (Xiao et al https://aclanthology.org/2023.emnlp-main.487.pdf) also calls their method HyperLoRA? Not sure how to reconcile this naming overlap.

---

> ### Author Response · Authors · 2024-11-22
>
> We thank the reviewer for the overall positive sentiment towards the paper and their thoughtful feedback. In particular, we agree with the reviewer that our proposed method provides a **simple but powerful tool to adapt LLMs without significant computational overhead**. We are glad to hear that they enjoy our **thorough ablations and clean presentation**.
>
> We would like to take the opportunity to discuss their suggestions and questions.
>
> > 1. Possible scaling concerns -- I find it a bit concerning that adding more training tasks doesn't improve the performance (Table 3). Usually for most algorithms, adding more data would result in a better model. Otherwise, the concern is that the performance of the method will be capped at a certain level and it will be hard to increase further.
>
> We fully agree with the reviewer that this is an important point. After the initial submission, we discovered that our HyperLoRA is undertrained in most cases. Thus, we have split the scaling experiment into two sections (**Section 5.1 and 5.2 in the revised version**). In **Section 5.1**, we scale the number of training tasks with a fixed compute budget across dataset sizes. HyperLoRA [L] still experiences performance degradation after 256 tasks with a fixed training budget. In **Section 5.2**, we scale the number of SGD steps proportional to the number of training tasks. After extended training, HyperLoRA [L] benefits from the additional training tasks and, thus, scales more gracefully. Additionally, HyperLoRA [M] improves over training runs with a fixed budget when using 256 or more training tasks.
>
> We hope this additional experiment addresses the reviewer’s concern regarding the scalability of the proposed method.
>
> > Also, in the paper, the model was evaluated on 11 different tasks (which are somewhat close to each other). I am wondering how the model would scale to even more tasks beyond the basic ones.
>
> We believe that the ten benchmarks included are representative of today’s LLM evaluation suite and are being used widely to benchmark frontier LLMs. Specifically, HyperLoRA is trained on SNI datasets and zero-shot evaluated on out-of-domain benchmark tasks (e.g., GSM8K, Humaneval, MBPP, BoolQ). These benchmarks are actively being used to evaluate modern frontier LLMs. They collectively evaluate different knowledge and skills of LLMs ranging from math, coding, reasoning, common sense, general knowledge, science knowledge, etc.
>
> Are there specific benchmarks that the reviewer wishes to see included? We are happy to include suggested benchmarks in the paper if time permits.
>
> > 2. Comparison with full fine-tuning -- The study is limited to the LoRA setting, which has been shown to perform worse than full fine-tuning, and this method likely wouldn't generalize to full fine-tuning settings. Similar to the above point, this makes me slightly concerned that the method might have its performance capped at a certain level.
>
> We thank the reviewer for this suggestion. Fine-tuning with LoRA is usually on par, in terms of performance, with full fine-tuning in many cases while being much more parameter efficient [1,2,3]. In theory, the hypernetwork could learn to generate delta for the pre-trained weights in the full weight space but would be much harder to train in practice (stability, VRAM requirement, training time, etc.) as the number of parameters scales with the size of the output space. In contrast, we believe an even more compact output space might allow the hypernetwork to better generalize and learn from diverse tasks. We leave this investigation for future work.
>
> Also, we have tried running full fine-tuning on Mistral-7B-Instruct but encountered the catastrophic forgetting problem, i.e., significantly degraded performance on various unseen benchmarks. We think that full fine-tuning requires a different training configuration. We will add the full fine-tuning baseline with proper training configuration in the camera-ready version.

---

> > ### Author Response · Authors · 2024-11-22
> >
> > > The paper claims to introduce HyperLoRA but one of the papers cited in the paper (Xiao et al https://aclanthology.org/2023.emnlp-main.487.pdf) also calls their method HyperLoRA? Not sure how to reconcile this naming overlap.
> >
> > Given the vast amount of work published in this area, we are sorry that we did not find a unique name before submission. We have changed the title of the paper to ***“Text-to-LoRA: Instant Transformer Adaption”***. We will contact the AC to change the title in Openreview accordingly.
> >
> > We thank the reviewer again for spending their time to provide constructive feedback. We hope to have addressed the reviewer’s remaining concerns and would highly appreciate it if the reviewer could confirm that the concerns have been addressed. If so, we’d like to ask the reviewer to consider increasing the recommendation score. We are happy to discuss further if any additional questions arise. We are looking forward to your response.
> >
> > Best wishes,
> >
> > The authors
> >
> > - [1] LoRA: Low-Rank Adaptation of Large Language Models. https://openreview.net/forum?id=nZeVKeeFYf9
> > - [2] Parameter-Efficient Fine-Tuning Methods for Pretrained Language Models: A Critical Review and Assessment. https://arxiv.org/pdf/2312.12148
> > - [3] VeRA: Vector-based Random Matrix Adaptation. https://openreview.net/forum?id=NjNfLdxr3A

---

> > > ### Author Response · Authors · 2024-11-26
> > >
> > > Dear Reviewer tn5E,
> > >
> > > We wanted to kindly remind you that the discussion period for the review process is coming to an end soon. We kindly ask if the reviewer could consider whether our clarifications and additional experiments support an increase in their score.
> > > We are happy to discuss further if any additional questions arise. We are looking forward to your response.
> > >
> > > Thank you once again for your time and feedback.
> > >
> > >
> > > Best regards,
> > >
> > > Authors

---

> > > > ### Author Response · Authors · 2024-12-03
> > > >
> > > > Dear Reviewer tn5E,
> > > >
> > > > We would like to gently remind you that the rebuttal period for the review process will conclude soon. We truly appreciate the time and effort you've already dedicated to reviewing our submission. Consequently, we have significantly improved and revised the manuscript including new experiments, baselines, and clarifications (see our general response to all reviewers).
> > > >
> > > > We would appreciate it if the reviewer could confirm whether the concerns have been addressed. Additionally, if the reviewer's concerns have been addressed, we would like to ask the reviewer to revise the rating score accordingly.
> > > >
> > > > If the reviewer has any further questions or concerns regarding our clarifications or additional experiments, please don't hesitate to reach out. We are eager to address any remaining points that might influence your final assessment.
> > > > Thank you once again for your valuable feedback and your continued support.
> > > >
> > > > Best regards,
> > > >
> > > > Authors

---

### Author Response · Authors · 2024-11-22
**Response to all reviewers**

# General comments to all reviewers
*Because of the naming conflict, we have changed the title of the paper to “Text-to-LoRA: Instant Transformer Adaption” (T2L) to highlight the focus of our work on generating LoRAs in a zero-shot manner using text instructions. We will contact the AC to change the title in Openreview accordingly. However, to avoid confusion, we still refer to the proposed method as HyperLoRA during the rebuttal period.*

We thank all reviewers for their thoughtful comments. The manuscript has been significantly enhanced to address the points of concern. Here, we discuss the novelty of the work and highlight the major revisions and additions to our paper.

**Novelty of our work:** We agree with reviewer jXry that our work is definitely not the first to use hypernetworks to generate on-the-fly adaption for language models. Nonetheless, we are convinced that our work adds a unique novel perspective with various advantages. We would like to point out the distinctions of our work here:

- We use frontier instruction fine-tuned models (decoder only) as the base models, i.e., Mistral-7B-Instruct-v0.2, Gemma-2-2b-it and Llama-3.1-8B-Instruct (added in **Section 4.3**), as opposed to prior work [1,2,3] that heavily focuses on pre-trained encoder-decoder architectures, e.g., T5 [4] or BART [5].
- Our evaluation setup is much more challenging than [1,2,3]. Specifically, our HyperLoRA is trained on SNI datasets and zero-shot evaluated on out-of-domain benchmark tasks (e.g., GSM8K, Humaneval, MBPP, BoolQ). These benchmarks are actively being used to evaluate modern frontier LLMs. In contrast, prior work only evaluates their methods on held-out SNI tasks.
- [1,2,3] rely on initializing part of their hypernetworks from the base model to achieve good performance or stable training. Our work, on the other hand, is task-embedder agnostic and can freely change the task embedding model (**Section 5.4**).
- We also explore the use of reconstruction loss to distill the knowledge of task-specific LoRAs into the hypernetwork and compare that to SFT-trained hypernetwork. We empirically find that reconstruction-trained HyperLoRA cannot generalize well compared to SFT-trained one (**Section 5.6**). Furthermore, we provide compelling evidence as to why that is the case. Thus, this work provides crucial insight into the training regime of the hypernetwork for fast adaptation (**Section 6.1**).

**Additional and revised experiments:** We have added a substantial amount of new experiments to address the concerns of all reviewers.
- We found several data contamination concerns with regard to the Super-NaturalInstructions (SNI) datasets and have corrected and revised the original SFT results after cleaning up the datasets (**Table 2,3,4,5,6,7,8,10**). HyperLoRA still consistently outperforms the baselines.
- Given the concerns about scaling (**reviewer tn5E**), we have conducted several revised and additional experiments: e.g. comparing dataset sizes (**Section 5.1**) and training compute (**Section 5.2**). We found that our original experiments significantly undertrained the HyperLoRA when increasing the number of datasets. Furthermore, we removed the dataset size cap (originally was capped to 500 samples per task). After extending the training, we find that HyperLoRA scales much more gracefully (Table 6). For a fair comparison, we also extend the training for multi-task LoRA.
- Furthermore, with extended training, we no longer use a prediction offset to kickstart the HyperLoRA (**reviewer M5pM**). Hence, learning the HyperLoRA end-to-end is beneficial and alleviates the need for a pre-trained multi-task LoRA offset.
- We have added a new section (**Section 4.3**), including experiments for two additional base models: Gemma-2-2b-it and Llama-3.1-8B-Instruct (**reviewer M5pM**).
- We have added a Hyperdecoders baseline (**reviewer jXry**), which generates the LoRA on a per-sequence basis (**Table 2**). Our HyperLoRA method compares favorably.
- We have added few-shot in-context learning (**reviewer M5pM**) and task description prepending (**reviewer SoVX**) as baselines (**Table 2,3,4**).
- As one of our distinctions is the use of generated task descriptions, we provide an additional ablation on the training description sources (generated vs SNI task definition) (**Section 5.3**)
- We have tried full fine-tuning on Mistral-7B-Instruct (**reviewer tn5E**) but encountered the catastrophic forgetting problem, i.e., significantly degraded performance on various unseen benchmarks. We think that full fine-tuning requires a different training configuration. We will add the full fine-tuning baseline with proper training configuration in the camera-ready version.

***All of our additional results are used to update the main text and we summarize the new figures*** [here](https://docs.google.com/presentation/d/e/2PACX-1vRo-SxPR7AXG9gQ28-Weaam5vDm3yJ7daa_wvytKp5rbW-PordB2D5ODuyM73WNYg7ynMF26Db-1eWd/pub?start=false).

---

> ### Author Response · Authors · 2024-11-22
> **Response to all reviewers (cont.)**
>
> We have removed typos, improved the presentation and clarity of the paper (**reviewer SoVX**) in various places, e.g., Figure 1 and Figure 3, and provided additional clarification of the experimental setup (**reviewer jXry**). Furthermore, we have removed all mentions of the prediction offset from the paper as HyperLoRA performs better without one under extended training. Thus, we hope that this will resolve any confusion around prediction offset (**reviewer SoVX and M5pM**).
>
> We excuse that the revised manuscript has an extra page due to significant improvements to the work. We will adjust the text to fit the page limit in the camera-ready version.
>
> Again, we thank the reviewers for significantly improving the work and hope these revisions could address the reviewers' concerns. We would appreciate it if the reviewers could check the updates in the paper (highlighted in brown). We are happy to further discuss and clarify if any reviewers feel their comments are not addressed. We are looking forward to your response.
>
>
> ### References
> - [1] HyperTuning: Toward Adapting Large Language Models without Back-propagation https://arxiv.org/abs/2211.12485
> - [2] Boosting Natural Language Generation from Instructions with Meta-Learning https://arxiv.org/abs/2210.11617
> - [3] HINT https://arxiv.org/pdf/2212.10315
> - [4] Exploring the Limits of Transfer Learning with a Unified Text-to-Text Transformer. https://arxiv.org/pdf/1910.10683
> - [5] BART: Denoising Sequence-to-Sequence Pre-training for Natural Language Generation, Translation, and Comprehension https://arxiv.org/abs/1910.13461

---

### Meta-Review · Area_Chair_L59L · 2024-12-25

**Metareview:**

This paper introduces Text-to-LoRA, proposing a hypernetwork-based approach for generating LoRA adapters through natural language task descriptions. The method claims to achieve comparable or superior performance to task-specific LoRAs across multiple model families (Mistral-7B, Gemma-2B, Llama-8B) without requiring per-task training. While the authors demonstrate empirical results across several benchmarks, the fundamental approach bears significant similarity to existing hypernetwork methods for adapter generation, particularly prior work like HyperTuning and HINT, with the main distinction being the application to different model architectures and evaluation settings.

The work shows competent empirical validation and practical implementation, with thorough ablation studies and clear experimental protocols. However, it suffers from critical limitations: inadequate theoretical novelty beyond existing hypernetwork approaches, unclear advantages over simpler baselines like in-context learning when accounting for computational costs, and limited theoretical analysis explaining why the approach works. Despite substantial revisions during rebuttal, the core contribution remains incremental, primarily applying known techniques to new architectures without introducing significant methodological innovations.  Thus I vote to reject this submission.

**Additional Comments On Reviewer Discussion:**

The rebuttal period revealed several fundamental issues with the submission. First, multiple reviewers identified significant overlap with prior hypernetwork approaches that wasn't adequately acknowledged in the initial submission. While the authors added comparisons and attempted to clarify distinctions, the core methodology remains highly similar to existing work, with the main differences being in implementation details and choice of base models. Second, concerns about experimental validation exposed limitations in the method's advantages - while the authors added new model families and baselines, the results showed minimal improvements over simpler approaches in many cases, and some baseline configurations appeared suboptimal, questioning the fairness of comparisons.

---

### Decision · Program_Chairs · 2025-01-22

Reject